# MMAB promotes negative feedback control of cholesterol homeostasis

Leigh Goedeke[1,2], Alberto Canfrán-Duque [1,3], Noemi Rotllan[1,3], Balkrishna Chaube [1,3], Bonne M. Thompson[4], Richard G. Lee [5], Gary W. Cline[2], Jeffrey G. McDonald [4], Gerald I. Shulman [2,6], Miguel A. Lasunción[7], Yajaira Suárez[1,3] & Carlos Fernández-Hernando [1,3✉]

Intricate regulatory networks govern the net balance of cholesterol biosynthesis, uptake and efflux; however, the mechanisms surrounding cholesterol homeostasis remain incompletely understood. Here, we develop an integrative genomic strategy to detect regulators of LDLR activity and identify 250 genes whose knockdown affects LDL-cholesterol uptake and whose expression is modulated by intracellular cholesterol levels in human hepatic cells. From these hits, we focus on *MMAB*, an enzyme which catalyzes the conversion of vitamin $B_{12}$ to adenosylcobalamin, and whose expression has previously been linked with altered levels of circulating cholesterol in humans. We demonstrate that hepatic levels of MMAB are modulated by dietary and cellular cholesterol levels through SREBP2, the master transcriptional regulator of cholesterol homeostasis. Knockdown of MMAB decreases intracellular cholesterol levels and augments SREBP2-mediated gene expression and LDL-cholesterol uptake in human and mouse hepatic cell lines. Reductions in total sterol content were attributed to increased intracellular levels of propionic and methylmalonic acid and subsequent inhibition of HMGCR activity and cholesterol biosynthesis. Moreover, mice treated with antisense inhibitors of MMAB display a significant reduction in hepatic HMGCR activity, hepatic sterol content and increased expression of SREBP2-mediated genes. Collectively, these findings reveal an unexpected role for the adenosylcobalamin pathway in regulating LDLR expression and identify MMAB as an additional control point by which cholesterol biosynthesis is regulated by its end product.

[1] Vascular Biology and Therapeutics Program, Yale School of Medicine, New Haven, CT, USA. [2] Department of Internal Medicine, Yale School of Medicine, New Haven, CT, USA. [3] Integrative Cell Signaling and Neurobiology of Metabolism Program, Department of Comparative Medicine and Pathology, Yale School of Medicine, New Haven, CT 06520, USA. [4] Center for Human Nutrition, University of Texas Southwestern Medical Center, Dallas, TX 75390, USA. [5] Cardiovascular Group, Antisense Drug Discovery, Ionis Pharmaceuticals, Carlsbad, CA 92010, USA. [6] Department of Cellular & Molecular Physiology, Yale School of Medicine, New Haven, CT, USA. [7] Servicio de Bioquímica-Investigación, Hospital Universitario Ramón y Cajal, Instituto Ramón y Cajal de Investigación Sanitaria (IRyCIS) and CIBER de Fisiopatología de la Obesidad y Nutrición (CIBERobn), Madrid, Spain. ✉email: carlos.fernandez@yale.edu

Cholesterol is an essential component of eukaryotic cells where it plays key structural and functional roles in a variety of pathways. Constituting around 60–90% of the lipid molecules in the plasma membrane, cholesterol serves to modulate cellular polarization and signal transduction[1]. In contrast, only 0.5–1% of total cellular cholesterol is present in the ER membrane, the site of cholesterol biosynthesis and sterol homeostatic machinery[2]. Cholesterol can also be found at intermediate levels in the mitochondria and peroxisomes, where it is oxidized and converted to steroids and bile acids to facilitate dietary lipid absorption[3]. Abnormal levels of cholesterol, however, can have serious cellular consequences, as evidenced by the excess cholesterol that accumulates in arterial walls to potentiate the development of atherosclerosis[4]. Therefore, mammals have developed complex mechanisms to regulate the abundance and distribution of sterols both at a systemic and cellular level.

The liver is a key organ involved in the regulation of cholesterol homeostasis and acquires cholesterol from de novo synthesis and from all classes of plasma lipoproteins. The synthesis of cholesterol occurs in the ER from acetyl-CoA through the mevalonate pathway[5]. The rate-limiting enzyme of this pathway is 3-hydroxy-3-methylglutaryl coenzyme A reductase (HMGCR), which catalyzes the conversion of HMG-CoA to mevalonic acid and is a common target for the cholesterol-lowering drugs, statins[6]. In addition to being synthesized, cholesterol can also be obtained from the circulation through the low density lipoprotein receptor (LDLR) in a classic example of receptor-mediated endocytosis[7]. Upon receptor binding and internalization, LDL is transported to the late endosomal/lysosomal system where LDL-cholesterol (LDL-C) esters are hydrolyzed by acid lipase (AL)[8]. Free cholesterol then exits the late endosome to downstream organelles through the coordinated action of the Niemann−Pick disease, type C1 and 2 (NPC1 and NPC2) proteins[9]. Other sterol-binding proteins are also involved in late endosomal trafficking; however, the mechanisms by which these proteins transport cholesterol are not well characterized[8,10,11].

The expression of the LDLR and other genes involved in cholesterol homeostasis are primarily governed by the coordinated actions of two transcription factor families: the liver X receptors (LXRs) and sterol regulatory element-binding proteins (SREBPs)[6,12,13]. When intracellular levels of cholesterol are high, oxidized derivatives of cholesterol activate LXRs to induce the expression of genes whose main function is to reduce intracellular cholesterol, including the cholesterol efflux pumps, ABCA1 and ABGC1, and the E3 ligase IDOL (the inducible degrader of the LDLR)[14]. In this setting, the ER-bound SREBPs also coordinate the transcriptional downregulation of LDLR, as well as the full set of genes involved in cholesterol biosynthesis[12]. In contrast, when cellular cholesterol levels decline, SREBPs induce the expression of HMGCR and LDLR, thereby increasing cholesterol biosynthesis, enhancing LDL clearance from the plasma and ensuring that cholesterol levels are maintained[13].

Components of the LXR and SREBP pathways are also subject to post-transcriptional regulation including the ubiquitination and degradation of HMGCR and SREBP[15] and the miRNA-mediated repression of SREBP2, HMGCR, and LDLR[16]. In particular, GWAS[17] and mouse studies[18,19] identified the proprotein convertase subtilisin/kexin type 9 (PCSK9) as a major regulator of plasma LDL-C levels by inducing the degradation of LDLR. This important finding has led to the development of anti-PCSK9 antibodies for treating individuals with high plasma LDL-C[20] and, together with the genes identified in recent GWAS, demonstrates that the regulation of hepatic LDLR expression and activity is complex and requires further investigation. As such, here, we use an integrative genomic strategy approach to identify the regulators of LDLR activity and cholesterol metabolism.

Specifically, we describe a role for MMAB, a SREBP2-target gene that catalyzes the conversion of vitamin $B_{12}$ to adenosylcobalamin, a cofactor necessary for the breakdown of certain amino acids, fatty acids, and cholesterol. By altering the levels of propionic and methylmalonic acid, loss of MMAB inhibits HMGCR activity and cholesterol biosynthesis, thereby increasing SREBP2-mediated gene expression and LDL-C uptake. Taken together, these findings identify an additional control point by which cholesterol biosynthesis is regulated by its end product.

## Results

**Genome-wide RNAi screen and expression profiling in human hepatic cells.** We recently developed a high-throughput microscope-based screening assay that monitored the effect of miRNA overexpression on the cellular internalization of 3′,3′-dioctadecylin-docarbocyanine-LDL (DiI-LDL) in human hepatic (Huh7) cells[21]. To identify protein-coding regulators of LDLR activity, we employed an integrative genomic strategy applying high-throughput RNA interference (RNAi) to our previously developed screening assay with genome-wide expression profiling in Huh7 cells (Fig. 1a). In order to establish the optimal conditions needed for identification of genes involved in LDL-C uptake and trafficking, we initially characterized SREBP2-mediated gene expression in Huh7 cells incubated in 10% lipoprotein-deficient serum (LPDS) and 30 μg/ml native LDL (nLDL) (Supplementary Fig. 1a). In agreement with the known regulatory functions of SREBP2 [22], addition of 30 μg/ml nLDL attenuated the expression of *LDLR* and *HMGCR*, the rate-limiting enzyme of cholesterol biosynthesis. Specifically, LDLR protein expression was significantly reduced at 8 h (Supplementary Fig. 1a), suggesting that LDL-C had reached the ER. Notably, the disruption of cholesterol exit from the late endosomes by the compound U18666A (U18) significantly increased *LDLR* and *HMGCR* mRNA expression. Moreover, the treatment of Huh7 cells with an siRNA directed against the endolysosomal protein, NPC1 (siNPC1), increased DiI-LDL uptake by 30% after 8 h (Supplementary Fig. 1b–c), suggesting our primary screen was suitable for the identification of proteins required for not only LDL-C uptake, but intracellular cholesterol transport, as well.

For systematic RNAi knockdown, Huh7 cells were plated in 384-well plates and reverse transfected with siRNAs against ~20,000 genes (Ambion, Silencer Human Genome siRNA Library V3) for 48 h (Fig. 1a). Previously validated siRNAs against NPC1 and LDLR, as well as a non-silencing (NS) siRNA, were used as positive and negative controls, respectively (Supplementary Fig. 1b–c). Each candidate gene was represented by a pool of three different siRNAs in triplicate. For functional characterization of DiI-LDL uptake, and thus LDLR activity, cells were incubated with 30 μg/ml DiI-LDL for an additional 8 h. Mean DiI-LDL intensity was determined at the level of single cells in five fields per well using quantitative image analysis software. Phenotypic effects caused by siRNAs were converted to robust z-scores based on the median average intensity of each respective array (includes randomly distributed NS control siRNAs) (Supplementary Fig. 1c). Upon normalization, robust z-scores for each individual siRNA were ranked and compared to respective replicate plates (Supplementary Fig. 1d). Only those genes were considered putative regulators of LDLR activity for which two or more replicate siRNAs yielded activities larger or smaller than 2 MAD away from non-silencing controls (deviation ≥ 2.0 or deviation ≤ −2.0) (Fig. 1c). Including controls, 1491 genes fulfilled these statistical criteria (Supplementary Data 1), validating loss of function of these genes as physiologically relevant in regulating LDL-C uptake, and thus receptor activity.

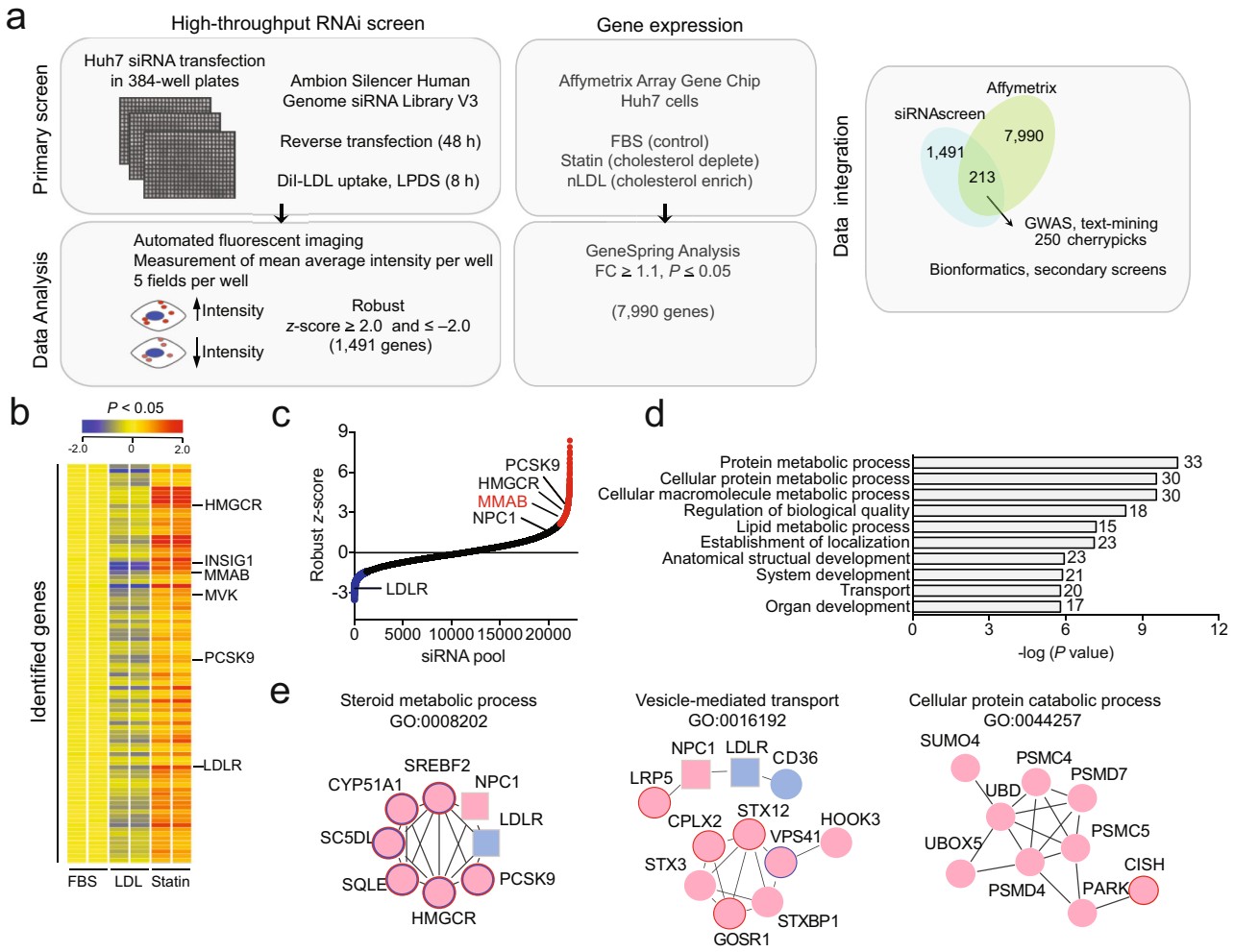

**Fig. 1 Genome-wide RNAi screen reveals regulators of LDLR activity. a** Outline of screening and bioinformatic procedures. **b** Gene expression analysis in Huh7 cells cultured in 10% fetal bovine serum (FBS, control), 120 µg/ml native LDL (nLDL, cholesterol enriched) or 5 µM statin (statin, cholesterol-depleted) for 24 h. Representative heatmap shows up- and downregulated genes (FC ≥ 1.1 in statin-treated cells compared to control; FC ≤ −1.1 in nLDL-treated cells compared to control; $P \le 0.05$, two-sided unpaired Student's $t$ test). Samples were analyzed in duplicate. Positive control genes are highlighted on the right. Also see Supplementary Data 2. **c** Distribution of average robust $z$-scores for individual siRNAs in the primary RNAi screen. siRNA pools corresponding to 22,203 genes were analyzed in triplicate. Positive (NPC1) and negative (LDLR) control genes, as well as known regulators of cholesterol metabolism (HMGCR, PCSK9) are highlighted. Red (robust $z$-score ≥ 2.0) and blue (robust $z$-score ≤ −2.0) siRNAs were chosen for further validation based on criteria in (**a**). All other siRNAs are shown in black. NS, non-silencing siRNA. **d** Negative log ($P$ values) of the top ten enriched GO terms identified in the primary RNAi screen. Gene set enrichment analysis was performed on the top 250 "cherry-picked" hits using MSigDB v5.0 (FDR $q$ value = 0.001). Number of genes in overlap ($k$) are depicted to the right of each GO term. **e** Interactions among hits associated with steroid metabolic process (left panel), vesicle-mediated transport (middle panel) and cellular protein catabolic process (right panel), as assessed using the STRING interaction database and Cytoscape plugin and outlined in Supplementary Data 1. Red circle, primary hit whose knockdown increased Dil-LDL uptake; blue circle, primary hit whose knockdown decreased Dil-LDL uptake. Positive and negative controls are depicted as squares. Colored border indicates hits also identified in Affymetrix array (**b**) whose expression decreased with nLDL treatment (blue) and/or increased with statin treatment (red). Source data are provided as a Source Data file.

In addition to our primary RNAi screen, we also performed unbiased genome-wide expression profiling in Huh7 cells cultured in media containing 10% fetal bovine serum (FBS) and 120 µg/ml nLDL (to cholesterol enrich) or 5 µM statin (to cholesterol deplete) for 24 h (Fig. 1b and Supplementary Data 2). Genes regulated at the transcriptional level by cellular cholesterol enrichment or depletion were then determined using an Affymetrix expression array. In total, the expression of 7990 genes were significantly decreased with nLDL treatment (fold-change ≥ 1.1, $P \le 0.05$) and/or increased with statin treatment (fold-change ≤ 1.1, $P \le 0.05$) (Fig. 1b). Among these, 25 genes were previously associated with the cholesterol metabolic process (GO:0008203), including *SREBP2, PCSK9, LDLR, INSIG1,* and

*NPC1* (Supplementary Data 2). Other genes whose expression was significantly altered with statin and LDL treatment included those of the cobalamin metabolic process (GO:0009235), innate immunity (GO:0045087), mitophagy (GO:0000422), NADP metabolic process (GO:0006739), cell proliferation (GO:0042127), gene expression regulation (GO:0010629), and receptor-mediated endocytosis (GO:0048261) (Supplementary Data 2).

**Analysis and validation of primary screening results.** To narrow down candidate regulators of cholesterol homeostasis, data obtained from gene expression profiling were integrated with results from our functional RNAi screen and compared to

genome-wide association studies (GWAS) associated with altered blood lipid levels. This initial comparison identified 250 primary hits, each of which whose knockdown significantly increased or decreased LDLR activity in at least two replicate screening plates and whose expression was modulated by intracellular cholesterol levels and/or associated with altered blood lipid levels and cardiovascular disease (CVD) in humans (Fig. 1a and Supplementary Data 1). Gene ontology (GO) enrichment analysis revealed that our dataset was markedly enriched in gene categories associated with protein metabolism, lipid metabolism, localization, and transport (Fig. 1d, Supplementary Fig. 1e, and Supplementary Data 1), which comprise functional categories previously associated with regulating LDLR activity. Further in-depth bioinformatic analysis of selected enriched functional categories using the STRING database revealed numerous interactions between factors associated within the same GO term (Figs. 1e and 2a). Specifically, we found multiple factors connected with the steroid metabolic process, including genes associated with the cholesterol biosynthetic pathway (HMGCR) and the post-transcriptional regulation of LDLR (PCSK9). Additionally, our analysis revealed several interactions among hits associated with vesicle-mediated transport, many of which have not been previously associated with regulating LDL-C uptake (Fig. 1e and Supplementary Data 1). Intriguingly, we also found candidate interactions associated with protein catabolism (Figs. 1e, 2a and Supplementary Data 1). Given the importance of the proteasome system in regulating cholesterol homeostasis[15], further analysis of these ubiquitin ligases is likely to identify regulators of LDLR activity in the future. Taken together, these analyses validate our primary screens as functionally relevant to identify regulators of cholesterol metabolism.

Next, to independently validate the significance of all 250 primary hits in regulating LDLR activity we performed a deconvolution screen. For each candidate gene, three individual siRNAs from the original siRNA pool were rescreened and tested for their ability to alter DiI-LDL uptake (Supplementary Fig. 2a–d and Supplementary Data 3). In total, we found that 160 hits produced a similar phenotypic effect observed in the primary screen (robust z score ≥ 1.6 or deviation ≤ −1.6) with at least 1 siRNA, comprising an overall validation rate of 64% (Supplementary Fig. 2d and Supplementary Data 3). From these 160 candidates, we selected 6 representative genes covering a broad range of GO processes (cobalamin biosynthetic process, protein ubiquitination, monoamine transport, and vesicle-mediated transport) and validated them using an independent siRNA library from Dharmacon (Fig. 2c). Among these six genes, knockdown of SLC6A4, MMAB, UBOX5, and TMEM14A increased LDLR activity to a level analogous to the primary and deconvolution screens (robust z score ≥ 1.5) (Fig. 2c). Of these, TMEM14A and MMAB were highly expressed in human liver tissue (Supplementary Fig. 3), validating loss of function of these candidates as physiologically relevant in regulating hepatic LDLR activity.

**MMAB is a regulator of cholesterol metabolism.** From the top candidate genes, we were initially interested in the methylmalonic aciduria (cobalamin deficiency) cblB type (MMAB) gene. MMAB encodes for a protein that catalyzes the final step in the conversion of vitamin $B_{12}$ into adenosylcobalamin (AdoCbl), a coenzyme for methylmalonyl-CoA mutase[23]. Interestingly, this gene is located in the same locus as mevalonate kinase (MVK)[24], a key enzyme in the cholesterol biosynthetic pathway (Fig. 3a). Moreover, several genome-wide association studies (GWAS) have found that the MMAB/MVK loci influences plasma levels of LDL-C and HDL-C in humans[25–27]. These intriguing findings led us to

further investigate the role of MMAB in regulating cellular cholesterol metabolism and LDLR activity in vitro and in vivo.

qRT-PCR analysis confirmed that MMAB was transcriptionally altered in Huh7 cells incubated with statin or nLDL compared to cells incubated in media containing 10% LPDS, similarly to other known cholesterol-sensing genes (Fig. 3b), and consistent with previous observations[24]. Comparable results were obtained at the protein level (Fig. 3c, d). Additionally, we found that MMAB expression was significantly upregulated in the livers of mice fed a chow diet supplemented with statin (Fig. 3e), suggesting that its regulation by dietary cholesterol is evolutionarily conserved. Analysis of the genomic reference sequence for MMAB revealed a conserved SREBP2-binding site upstream of the MMAB promoter, as well as binding sites for generic transcription factors involved in SREBP activation, including SP1 (Fig. 3a). This motif is shared by MVK, and suggests that MMAB is regulated with MVK by SREBP2 as part of a bifunctional locus. Indeed, MMAB/MVK promoter activity was significantly increased when cells were cholesterol-depleted with statin treatment, similarly to LDLR (Fig. 3f). Consistent with this, overexpression of the nuclear, active form of SREBP2 (nSREBP2) dose-dependently increased MMAB and MVK promoter activity (Fig. 3g).

To further confirm that MMAB regulates LDLR activity in human hepatic cells, we next inhibited MMAB expression using an independent siRNA from Dharmacon and examined DiI-LDL uptake by fluorescence microscopy. As expected by the results obtained in the primary and secondary screening assays, the inhibition of MMAB markedly increased DiI-LDL uptake, compared to cells transfected with an NS control siRNA (Fig. 4a, upper panel). Similar results were observed when we assessed DiI-LDL binding and uptake by flow cytometry (Fig. 4b, lower panel). To begin to understand how MMAB controls LDL-C uptake, we next assessed the effect of MMAB knockdown on LDLR mRNA and protein expression. Transfection of Huh7 cells with an siRNA against MMAB (siMMAB), but not a control (NS) siRNA, significantly increased LDLR mRNA and protein expression (Fig. 4b, c). Interestingly, knockdown of MMAB also significantly upregulated the mRNA and protein expression of HMGCR, SREBP2, CYP51A, and PCSK9, all known targets of SREBP2 (Fig. 4b, c). Given that SREBP2 is classically activated by low levels of intracellular cholesterol, we hypothesized that MMAB may be controlling LDL-C uptake by reducing the levels of intracellular cholesterol, thereby increasing SREBP2-mediated gene expression, including that of LDLR. Indeed, when we assessed the total sterol content of Huh7 cells transfected with an siRNA against MMAB, total levels of intracellular cholesterol were significantly decreased (Fig. 4d). Intriguingly, upstream sterol intermediates were also significantly reduced, suggesting that knockdown of MMAB results in reduced intracellular cholesterol levels by altering cholesterol biosynthesis. Consistent with our hypothesis, cells transfected with siMMAB had reduced [14C]-acetate incorporation into cholesterol (Fig. 4e). Taken together, these data suggest that MMAB controls LDLR expression and activity by modulating cholesterol biosynthesis.

Next, we assessed whether suppression of MVK, an underlying susceptibility gene in the lipid-associated MMAB-MVK locus[25–27], also increased the expression and activity of LDLR in human hepatoma cells. In contrast to the results observed in MMAB-depleted cells, we found that knockdown of MVK levels markedly reduced LDLR expression and attenuated DiI-LDL binding and uptake (Supplementary Fig. 4a–d). To further characterize how MVK silencing affects LDLR expression, we transfected Huh7 cells with an siRNA against MVK (siMVK) or NS control siRNA and analyzed the expression of LDLR at different time points. Notably, we observed a marked

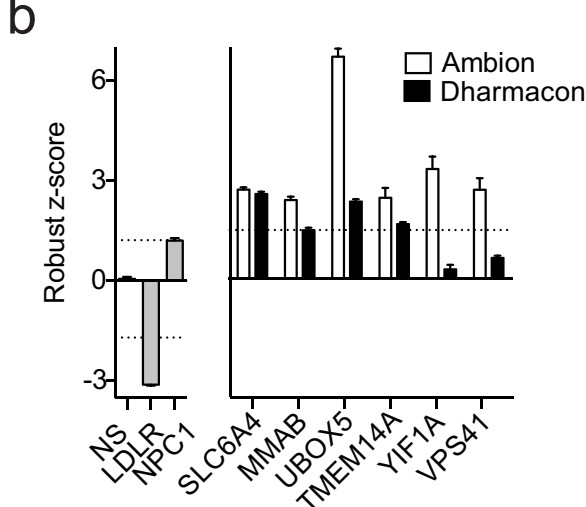

upregulation of LDLR 24 h after MVK knockdown and a significant decrease at later time points (Supplementary Fig. 4e), suggesting that mevalonate and/or other derivatives of mevalonic acid provide a signal for repression of LDLR expression. Indeed, previous studies have reported the inhibitory effect of mevalonolactone on hepatic LDLR expression in rats treated with Zaragozic acid, an inhibitor of squalene synthase[28].

**Increased LDLR activity in methylmalonic acidurias.** To explore the role of MMAB in regulating cholesterol biosynthesis, we next assessed the effect of MMAB deficiency on the global metabolic profile of Huh7 cells. Loss of MMAB function leads to reduced catabolism of odd-chain fatty acids, cholesterol, and the amino acids, valine, isoleucine, methionine, and threonine (Fig. 5a). This deficiency produces elevated levels of toxic organic

**Fig. 2 Primary RNAi screen validation. a** Functional map of top 250 hits from primary RNAi screen. Top hits identified in the primary screens (Fig. 1a) were uploaded into DAVID for functional annotation cluster analysis. Individual genes were assigned to the highest enriched functional annotation cluster in which they were present (see Supplementary Data 1). Regulators of LDLR activity in functional clusters that interact or share functional annotations are depicted in colored circles. Selected genes not found within functional annotation clusters are indicated as white hexagons. Lines between genes in the same and different functional clusters indicate STRING interaction score. Approximate cellular locations of each functional gene cluster are pictured. **b** Average robust z scores of top six hits validated in a secondary screening assay using siRNAs from independent libraries. Gray bars, positive and negative screening controls; white bars, Ambion siRNA library used in primary screen; black bars, Dharmacon siRNA library used in secondary screen. Data are mean ± s.e.m. $n = 3$ technical replicates for Ambion siRNA library and 12−32 technical replicates for Dharmacon siRNA library. Dashed line: robust z score = 1.5 or −1.5. Source data are provided as a Source Data file.

acids (propionic and methylmalonic acid) in patients with the autosomal recessive disorder methylmalonic aciduria[29,30]. As shown in Fig. 5b and c, knockdown of MMAB caused a marked accumulation of methylmalonic acid and propionylcarnitine, the carnitine-conjugated derivative of propionyl-CoA that is upstream of methylmalonyl-CoA and indicative of reduced methylmalonyl-CoA mutase (MUT) activity. Of note, other proximal metabolites, such as acetyl-CoA and succinyl-CoA, were unchanged (Supplementary Data 4).

If siMMAB-mediated alterations in cholesterol biosynthesis and LDLR activity were due to changes in metabolic inter-mediates, we posited that loss of other enzymes in this metabolic pathway would produce a similar phenotype. Indeed, cultured fibroblasts from patients deficient in the mitochondrial enzyme, *MUT*, accumulated significantly higher levels of propionylcarni-tine and other metabolites produced from the degradation of branched-chain amino acids (isobutyrylcarnitine, isovalerylcarni-tine, 2-methylbutyrylcarnitine) (Fig. 5d and Supplementary Data 5). Intriguingly, *MUT*-deficient fibroblasts also had markedly increased DiI-LDL uptake compared to age- and sex-matched healthy (WT fibroblasts) individuals (Fig. 5e). Consistent with this, the mRNA and protein levels of SREBP2-target genes, such as LDLR, were significantly increased (Fig. 5f, g). Furthermore, when we assessed the total sterol content of *MUT*-deficient fibroblasts by GC-MS, we found that intracellular cholesterol levels and other sterol intermediates were prominently reduced (Fig. 5h), suggesting that cholesterol homeostasis is altered in methylmalonic acidurias, and that metabolic changes may be contributing siMMAB-mediated reductions in cholesterol biosynthesis.

To further ascertain the importance of the adenosylcobalamin pathway in regulating LDLR activity, we next determined whether double inactivation of *MMAB* and *MUT* enhanced the expression of LDLR expression in human hepatoma cells. To this end, we silenced the expression of *MMAB*, *MUT* or both genes in Huh7 cells and evaluated the expression of LDLR. Similar to our previous results, we found a significant increase in LDLR protein expression in Huh7 cells treated with an siRNA against MUT or MMAB compared to cells transfected with an NS control siRNA (Supplementary Fig. 5a and b). Interestingly, the suppression of both genes led to a modest increase of LDLR expression compared to cells treated with either MMAB or MUT siRNA alone, suggesting a potential additional effect of both genes in regulating LDLR expression and cholesterol homeostasis (Supplementary Fig. 5a and b).

**Propionic acid increases LDLR expression and activity.** Previous reports have shown that propionic acid inhibits cholesterol biosynthesis in isolated rat hepatocytes[31]. Therefore, we hypothesized that the loss of MMAB may be affecting LDLR expression and activity by increasing intracellular concentrations of propionate. To test this idea, we initially treated Huh7 cells with varying doses of propionic acid for 24 h and assessed LDLR activity using FACS. Importantly, concentrations of propionic

acid (~75−100 μM) previously found in the portal vein of humans[32,33] and known to increase intracellular propionyl-CoA levels similarly to those found in primary human hepatocytes from patients with methylmalonic aciduria[34] significantly increased DiI-LDL specific uptake and binding (Supplementary Fig. 6a and Fig. 6a, b). Consistent with this, 100 μM propionic acid also markedly increased the expression of LDLR and other SREBP-responsive genes at the mRNA and protein level (Supplementary Fig. 6b and Fig. 6c). Given that SREBP2 is a crucial transcriptional regulator of LDLR[12], we next investigated whether SREBP2 was required for the propionate-mediated upregulation of LDLR. To this end, Huh7 cells were treated with the site 1 protease (S1P) inhibitor, PF-429242 (which inhibits endogenous SREBP processing[35]), and propionic acid or vehicle control. Importantly, in Huh7 cells incubated with PF-429242 (S1P Inh), the mRNA and protein levels of LDLR and other cholesterol-sensing genes (HMGCR, SREBP2, PCSK9, CYP51A, and DHCR24) were markedly suppressed (Supplementary Fig. 7a, b). Interestingly, the propionate-mediated upregulation of LDLR, HMGCR, CYP51A, and mature SREBP2 (mSREBP2) was abolished in cells treated with S1P Inh (Supplementary Fig. 7a, b), suggesting that propionate upregulates LDLR expression through the transcriptional activation of SREBP2. Because SREBP2 expression is induced by low intracellular cholesterol levels, we next assessed the total sterol content of Huh7 cells incubated with 100 μM propionic acid for 24 h. As seen in Supplementary Fig. 7c, propionic acid significantly reduced intracellular sterol levels, as well as [$^{14}$C]-acetate incorporation into cholesterol, similarly to what has previously been published[31]. Interestingly, the propionate-mediated reduction in cholesterol biosynthesis was not due to direct inhibition of HMGCR, as HMGCR activity was unaltered in isolated liver microsomes treated with physiologic and supraphysiologic doses of propionic acid (Fig. 5d). Taken together, these results suggest that siMMAB-mediated increases in propionate control cholesterol biosynthesis through an indirect mechanism.

**Methylmalonic acid upregulates LDLR activity by inhibiting HMGCR activity and reducing cholesterol biosynthesis.** Mutations in MMAB impair the activity of *MUT*, leading to the accumulation of toxic metabolites, including methylmalonic acid[29,30]. To determine whether increases in methylmalonic acid could also contribute to siMMAB-mediated increases in LDLR activity, we next assessed DiI-LDL-specific uptake in Huh7 cells treated with varying doses of methylmalonic acid for 24 h. As shown in Supplementary Fig. 6c, 25−200 μM methylmalonic acid (levels previously observed in primary human hepatocytes iso-lated from patients with methylmalonic aciduria[34] and in the serum of aged individuals[36]) significantly increased DiI-LDL-specific uptake compared to controls. Consistent with this, 50 μM methylmalonic acid significantly increased the expression of LDLR and other SREBP-responsive genes at the mRNA and protein level (Supplementary Fig. 6d and Fig. 6e). Importantly, when we assessed HMGCR activity in isolated hepatic

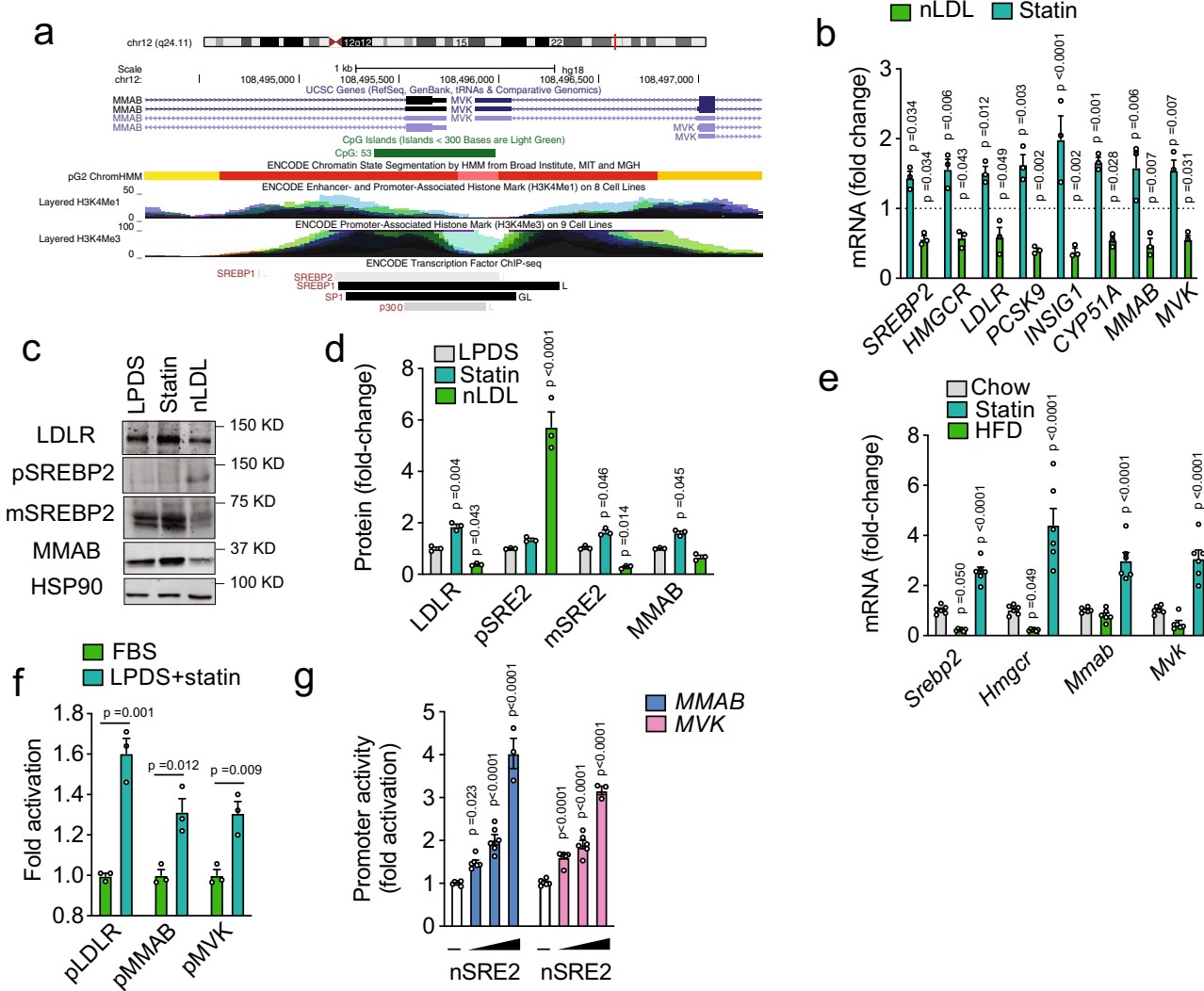

**Fig. 3 SREBP2 regulates MMAB/MVK expression in vitro and in vivo. a** Schematic diagram of human chromosome 12, showing the localization of *MMAB* and *MVK*. The active promoter region of *MMAB* and *MVK* is shown in red (HepG2 ChromHMM) and correlates with CpG islands (green) and enriched H3K4Me3 marks. Transcription factor binding sites (as assayed by ChIP-seq) are shown below. SREBP2, SREBP1, SP1, and p300 are highlighted in red. Data were compiled using the UCSC Genome Browser (NCBI36/hg18, http://genome.ucsc.edu). **b** qRT-PCR analysis of SREBP2-responsive genes in Huh7 cells incubated in lipoprotein-deficient serum (LPDS), LPDS + 5 µM statin, or LPDS + 120 µg/ml native LDL (nLDL) for 24 h. Relative expression levels were normalized to those cells incubated in LPDS (dashed line). Data are mean ± s.e.m. of three independent experiments in duplicate. Statistical comparisons by two-way ANOVA with Bonferroni correction for multiple comparisons. **c**, **d** Representative Western blot of LDLR, SREBP2, and MMAB in Huh7 cells incubated in lipoprotein-deficient serum (LPDS), LPDS + 5 µM statin, or LPDS + 120 µg/ml native LDL (nLDL) for 24 h. HSP90 was used as a loading control. Quantification of blot shown in panel (**d**). Data are mean ± s.e.m. of three independent experiments. Statistical comparisons by two-way ANOVA with Bonferroni correction for multiple comparisons. **e** qRT-PCR analysis of SREBP2-responsive genes in the livers of C57BL/6 mice fed a chow diet, chow diet supplemented with statin (statin) or high-fat diet (HFD). Data are mean ± s.e.m. n = 6 mice per group. Statistical comparisons by two-way ANOVA with Bonferroni correction for multiple comparisons. **f** *MMAB*, *MVK* and *LDLR* promoter activity in Huh7 cells incubated in 20% FBS + 60 µg/ml native LDL (nLDL) or LPDS and 5 µM statin for 24 h (LPDS + statin). Data are mean ± s.e.m. of three independent experiments in triplicate. Statistical comparisons by two-sided unpaired Student's *t* test. **g** *MMAB* and *MVK* promoter activity in HeLa cells transfected with an empty vector (−) or 0.5, 1, and 2 µg nuclear SREBP2 (nSRE2). Data are mean ± s.e.m. of six independent experiments in duplicate. Statistical comparisons by one-way ANOVA with Bonferroni correction for multiple comparisons. Source data are provided as a Source Data file.

microsomes, we found that methylmalonic acid significantly inhibited HMGCR activity at doses as low as 50 µM (Fig. 6f), consistent with methylmalonic acid mediating a direct effect of siMMAB-mediated reductions in cholesterol biosynthesis.

**MMAB negatively regulates LDLR expression in vivo by altering HMGCR activity and SREBP2-mediated gene expression.** To further evaluate the role of MMAB in regulating LDLR activity in vivo, we next silenced MMAB expression in mice using antisense oligonucleotides (ASOs) and antisense locked nucleic

acid (LNA) GapmeRs. Initially, we screened four different GapmeRs directed against *Mmab* mRNA (LNA#1−4) for their ability to silence MMAB in mouse hepatic (Hepa) cells. Transfection of Hepa cells with 50 nM of LNA #3 and #4 significantly reduced MMAB mRNA and protein expression compared to controls (Supplementary Fig. 8a–b). Moreover, LNA #3 (LNA MMAB) markedly increased DiI-LDL-specific uptake in Hepa cells, thus confirming our previous results using siRNAs in Huh7 cells and suggesting that the siMMAB-mediated regulation of LDLR expression is evolutionarily conserved (Supplementary Fig. 8c–d).

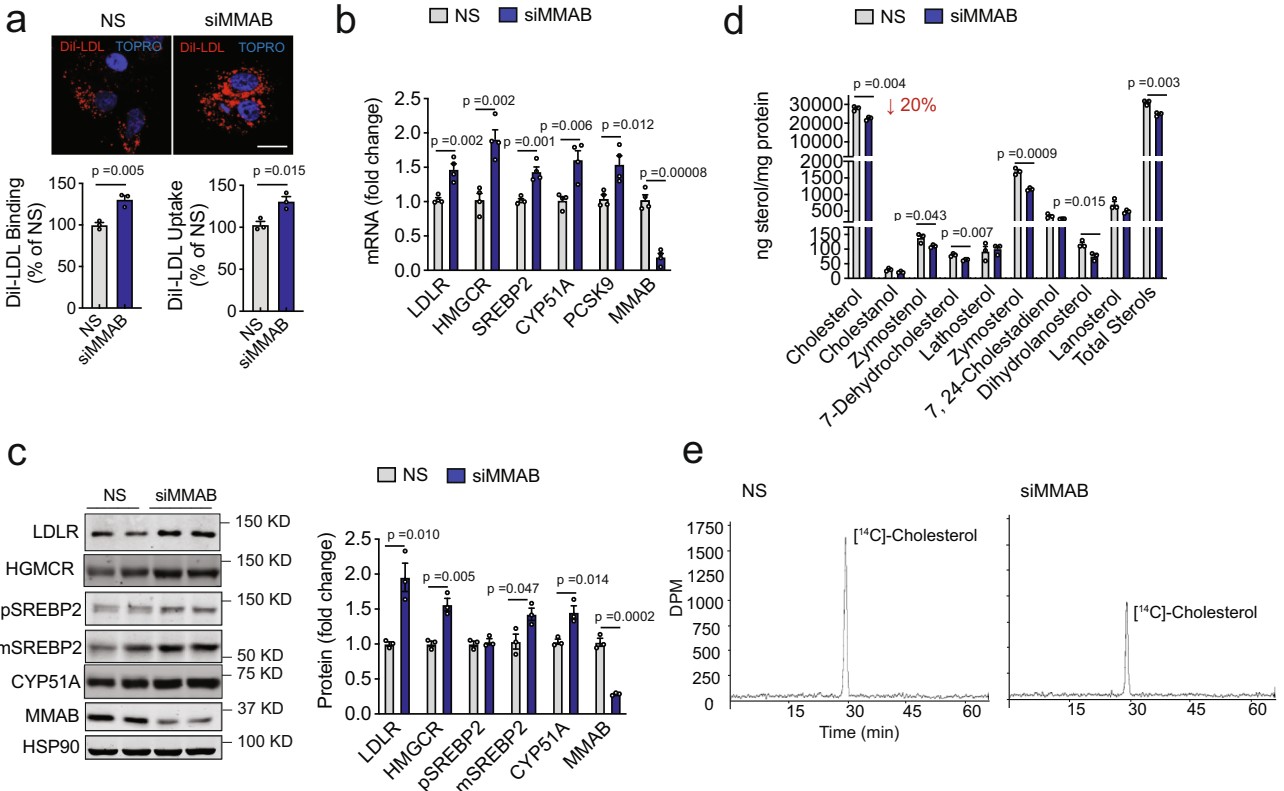

**Fig. 4 Knockdown of MMAB increases LDLR expression and activity and decreases cholesterol biosynthesis in vitro. a** Representative images of DiI-LDL uptake in Huh7 cells transfected with a non-silencing siRNA (NS) or siRNA MMAB (siMMAB) and incubated with 30 μg/ml DiI-LDL for 8 h at 37 °C. Red, DiI-LDL; blue, nuclei stained with TO-PRO-3 iodide (TOPRO). Scale bar, 10 μm. Flow cytometry analysis of DiI-LDL specific binding and uptake in Huh7 cells transfected with NS siRNA or siMMAB and incubated with 30 μg/ml DiI-LDL for 8 h at 37 °C (uptake) or 90 min at 4 °C (binding) is shown below. Data are mean ± s.e.m. of three independent experiments. Statistical comparisons by two-sided unpaired Student's $t$ test. **b, c** qRT-PCR (**b**) and Western analysis (**c**) of SREBP2-responsive genes in Huh7 cells transfected with an NS control siRNA or siMMAB. HSP90 was used as a loading control. Quantification compared to HSP90 is shown in panel (**c**). Data are mean ± s.e.m. of three (**c**) or four (**b**) independent experiments in duplicate. Statistical comparisons by two-sided unpaired Student's $t$ test. **d** Sterol content of Huh7 cells transfected with NS siRNA or siMMAB. Lipids were extracted and analyzed by GC-MS. "Total sterols" were calculated as the sum of all sterols that could be detected. Data are mean ± s.e.m. of three independent experiments. Statistical comparisons by two-sided unpaired Student's $t$ test. **e** Effect of MMAB knockdown on [1,2-$^{14}$C]-acetate incorporation into cholesterol as determined by HPLC. Huh7 cells were treated as in (**d**). Data are representative of three independent experiments. Source data are provided as a Source Data file.

To address the consequence of silencing MMAB in vivo, 8-week-old male mice were fed a chow diet and injected weekly with an antisense oligonucleotide against MMAB (MMAB ASO) or control oligonucleotide [(CON ASO), 50 mg/kg × 6 weeks] (Supplementary Fig. 8e). As shown in Fig. 7a and d, treatment with MMAB ASO led to a marked reduction in hepatic *Mmab* mRNA and protein levels, without altering *Mvk* expression. Consistent with this, plasma and liver MMA levels were significantly increased (Fig. 7b, c). Body weight, total plasma cholesterol, HDL-C, and transaminase levels were unaltered (Supplementary Fig. 8f–j).

To ascertain the mechanism by which loss of MMAB may affect LDLR expression in vivo, we next performed RNA-seq analysis in the livers of mice treated with MMAB ASO or CON ASO. Knockdown of MMAB significantly upregulated many SREBP2-responsive genes involved in the cholesterol biosynthetic and mevalonate pathways, including *Mvk* and *Hmgcr* (Fig. 7e–g and Supplementary Data 6). Similar results were obtained at the protein level (Fig. 7d). To determine whether the siMMAB-mediated increase in SREBP2-responsive genes was due to the inhibition of cholesterol biosynthesis, we next measured HMGCR activity in the livers of mice treated with MMAB or CON ASO. Consistent with our in vitro data, knockdown of MMAB

significantly reduced HMGCR activity (Fig. 7h), suggesting that MMAB may increase hepatic LDLR expression by negatively regulating cholesterol biosynthesis.

Increased expression of hepatic LDLR expression is considered beneficial for reducing plasma cholesterol and atherosclerosis. To assess whether loss of MMAB alters circulating plasma cholesterol, we next treated Western-diet fed *Apobec1*$^{-/-}$*Ldlr*$^{+/-}$ mice, which display a LDL-dominant lipoprotein profile[37,38], with 5 mg/kg LNA MMAB or a scrambled control LNA (LNA control) (Supplementary Fig. 8k) for 2 weeks. Twenty-four hours after the last injection, mice were sacrificed and plasma and livers collected for cholesterol and gene expression analysis. As shown in Supplementary Fig. 8l–n, treatment with LNA MMAB significantly decreased hepatic MMAB levels at the mRNA and protein level, without altering body weight. The protein levels of LDLR and other SREBP2-mediated genes, including CYP51A and DHCR24, were also slightly, but significantly increased (Supplementary Fig. 8n). Furthermore, knockdown of MMAB significantly reduced hepatic sterol content (Supplementary Fig. 8o), in accordance with our in vitro findings and the reduction in HMGCR activity found in wild-type mice treated with MMAB ASO (Fig. 7h). While MMAB LNA treatment significantly increased hepatic LDLR expression, total plasma cholesterol

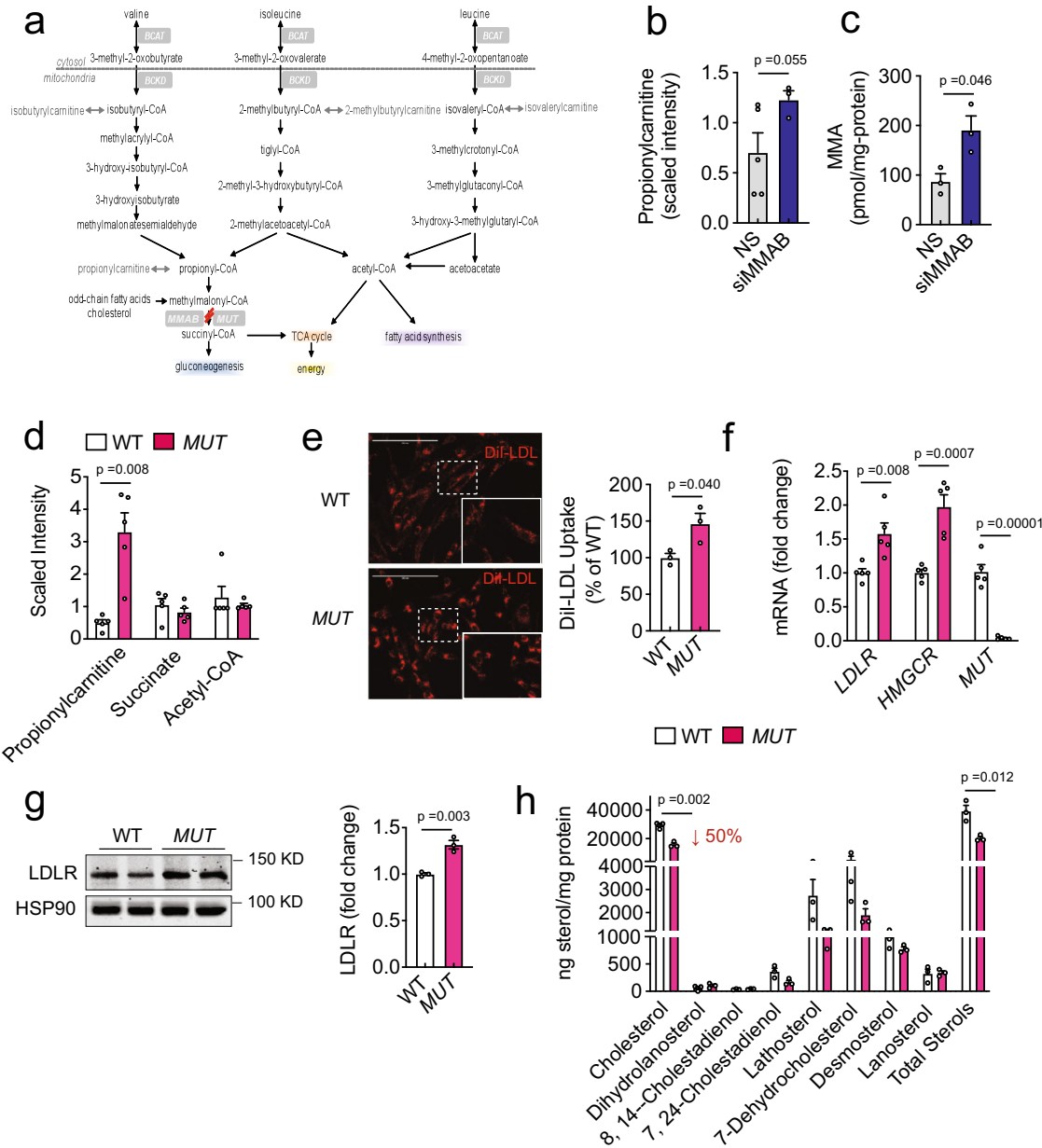

**Fig. 5 *MUT*-deficient human fibroblasts have increased LDLR expression and function and decreased sterol content. a** Catabolism of odd-chain fatty acids, cholesterol, and amino acids. *MMAB* catalyzes the final step in the conversion of vitamin B$_{12}$ into adenosylcobalamin (AdoCbl), a coenzyme for methylmalonyl-coA mutase (*MUT*). Mutations in *MMAB* or *MUT* lead to the metabolic disorder, methylmalonic aciduria. **b, c** Propionylcarntitine (**b**) and methylmalonic acid (MMA) (**c**) content in Huh7 cells transfected with a non-silencing siRNA (NS) or siRNA MMAB (siMMAB) for 60 h. Data are mean ± s.e.m. $n = 3$–5 biological replicates per group. Statistical comparisons by unpaired Welch's two-sample *t* test. **d** Relative levels of propionylcarnitine, succinate and acetyl-CoA in WT and *MUT*-deficient fibroblasts incubated in DMEM containing 10% LPDS for 24 h. Data are mean ± s.e.m. of five independent experiments. Statistical comparisons by unpaired Welch's two-sample *t* test. **e** Representative images of DiI-LDL uptake in human fibroblasts deficient of methylmalonyl-coA mutase (*MUT*) or wild-type (WT) controls and incubated with 30 µg/ml DiI-LDL for 2 h at 37 °C. Red, DiI-LDL. Scale, 200 µm. Quantification of DiI-LDL uptake by FACS is shown to the right. Data are mean ± s.e.m. of three independent experiments. Statistical comparisons by unpaired two-sided Student's *t* test. **f, g** qRT-PCR (**f**) and Western blot (**g**) analysis of SREBP2 responsive genes in WT and *MUT* fibroblasts incubated in LPDS for 24 h. HSP90 was used as a loading control. Quantification of blot is shown to the right. Data are mean ± s.e.m. of 3–5 independent experiments in duplicate, respectively. Statistical comparisons by unpaired two-sided Student's *t* test. **h** Sterol content of WT and *MUT* fibroblasts incubated in DMEM containing 10% LPDS for 24 h. Lipids were extracted and analyzed by GC-MS. "Total sterols" were calculated as the sum of all sterols that could be detected. Data are mean ± s.e.m. of three independent experiments. Statistical comparisons by unpaired two-sided Student's *t* test. Source data are provided as a Source Data file.

levels and HDL-C levels were unchanged (Supplementary Fig. 8p–q). When we analyzed fasting plasma lipid levels by FPLC, however, LDL-C levels tended to be reduced (Supplementary Fig. 8r). Collectively, these studies suggest that MMAB regulates LDLR expression and activity in vivo, most likely through the MMA-mediated inhibition of HMGCR activity, reduction in cholesterol biosynthesis and subsequent upregulation of SREBP2-mediated gene expression (Fig. 8a, b).

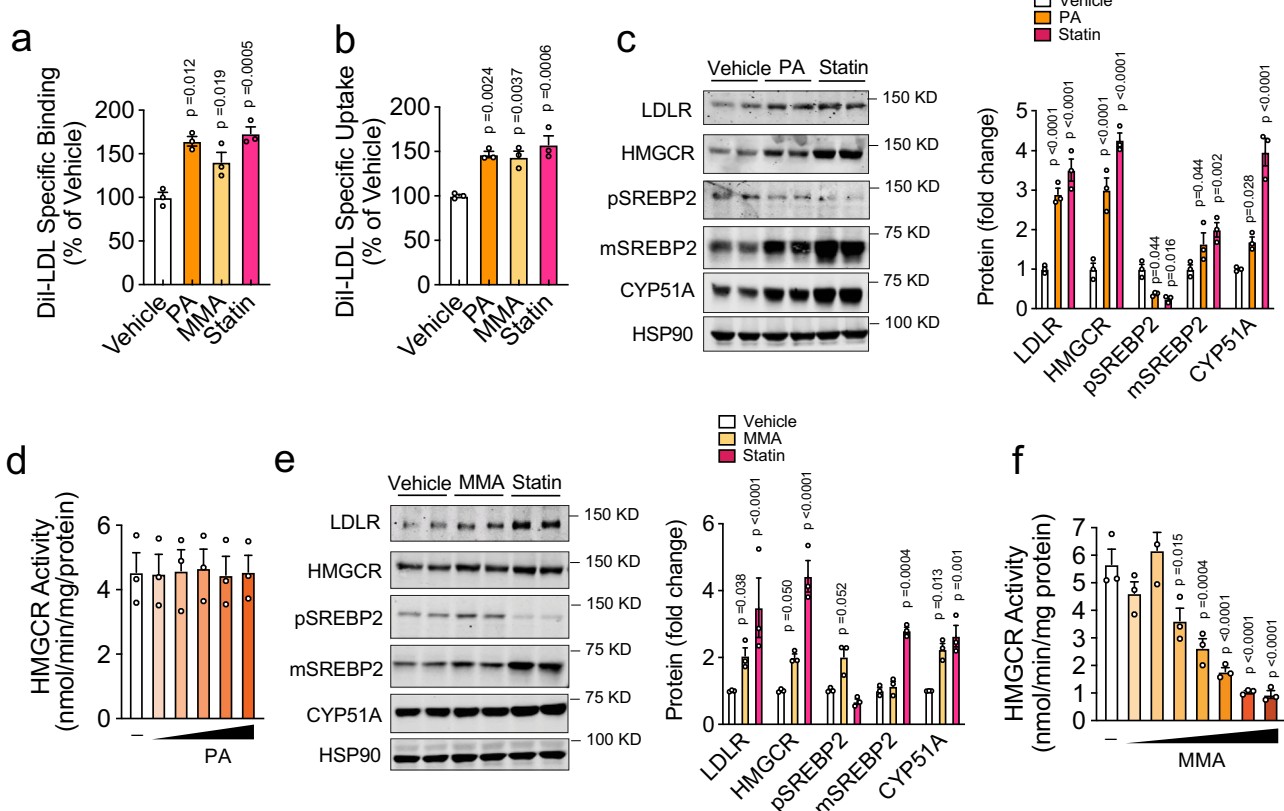

**Fig. 6 Methylmalonic acid inhibits HMGCR activity and increases SREBP2-mediated gene expression and DiI-LDL uptake. a, b** Flow cytometry analysis of DiI-LDL specific binding (**a**) and uptake (**b**) in Huh7 cells treated with vehicle, 50 μM methylmalonic acid (MMA), 100 μM propionic acid (PA) or 5 μM statin for 24 h and incubated with 30 μg/ml DiI-LDL for 2 h at 37 °C (uptake) or 90 min at 4 °C (binding). Data are mean ± s.e.m. of three independent experiments in duplicate. Statistical comparisons by one-way ANOVA with Bonferroni correction for multiple comparisons. **c** Western blot analysis of SREBP2-responsive genes in Huh7 cells incubated in LPDS for 24 h and treated with vehicle, 100 μM propionic acid (PA), or 5 μM statin for 24 h. HSP90 was used as a loading control; quantification is shown to the right of the blots. Data are mean ± s.e.m. of three independent experiments in duplicate. Statistical comparisons by two-way ANOVA with Bonferroni correction for multiple comparisons. **d** HMGCR activity in liver microsomes treated with increasing doses (10 μM, 100 μM, 1 mM, 10 mM, 100 mM) of propionic acid or vehicle control. Data are mean ± s.e.m. of three independent experiments in duplicate. **e** Western blot analysis of SREBP2-responsive genes in Huh7 cells incubated in LPDS for 24 h and treated with vehicle, 50 μM methylmalonic acid (MMA), or 5 μM statin for 24 h. HSP90 was used as a loading control; quantification is shown to the right of the blots. Data are mean ± s.e.m. of three independent experiments in duplicate. Statistical comparisons by two-way ANOVA with Bonferroni correction for multiple comparisons. **f** HMGCR activity in liver microsomes treated with increasing doses (10 μM, 25 μM, 50 μM, 75 μM, 100 μM, 1 mM, 10 mM) of methylmalonic acid (MMA) or vehicle control. Data are mean ± s.e.m. of three independent experiments in duplicate. Statistical comparisons by one-way ANOVA with Bonferroni correction for multiple comparisons.

## Discussion

While many significant advances have been made in understanding the balance between cholesterol synthesis and transport, several issues warrant further characterization. In particular, the physiological functions of various proteins involved in LDL-C trafficking and regulation have yet to be uncovered and there is an apparent need to study the orchestration of sterol balance in key cells that are relevant for whole-body cholesterol metabolism, particularly hepatocytes and enterocytes[8,10,11]. Moreover, the molecular interactions of already identified LDL-C regulating factors, as well as the subsequent events that direct cholesterol homeostasis on a functional level are poorly understood[39]. With the aim to comprehensively dissect the genes and pathways that regulate LDLR activity, we employed an unbiased, systemic functional genomic strategy that combined RNAi with our previously developed high-throughput screening assay for DiI-LDL-cholesterol uptake[21] and genome-wide expression profiling in Huh7 cells. Among the validated regulators of LDLR activity, we found enrichment for factors with established (HMGCR, SREBP2, PCSK9)[12,20] roles in regulating cellular cholesterol

homeostasis and LDLR activity. Intriguingly SNPs in *HMGCR* and *PCSK9* have previously been reported as causative for imbalanced LDL-C levels that predispose individuals to CVD[40]. Moreover, given that PCSK9 is a SREBP2-target gene[41], it is likely that knockdown of *SREBP2* led to increased DiI-LDL uptake by reducing PCSK9 expression.

Aside from the classical regulators of cholesterol homeostasis, our results confirm and extend previously less well-described regulators of LDLR activity. For example, knockdown of MAP3K1 (Supplementary Data 1) reduced LDLR activity, consistent with its role as a transcriptional activator of LDLR by SREBP2 [42]. Additionally, our RNAi screen identified COPB1 as a negative regulator of DiI-LDL-cholesterol uptake (Supplementary Data 1). COPB1 is a protein subunit of the coatomer complex and has previously been implicated in the processing, activity, and recycling of the LDLR[39]. Taken together, these results demonstrate that our genomic approach identified candidates with a direct role in regulating LDLR activity or indirect role by modulating cholesterol homeostasis through SREBP2. As such, although the molecular roles of the other top candidate genes

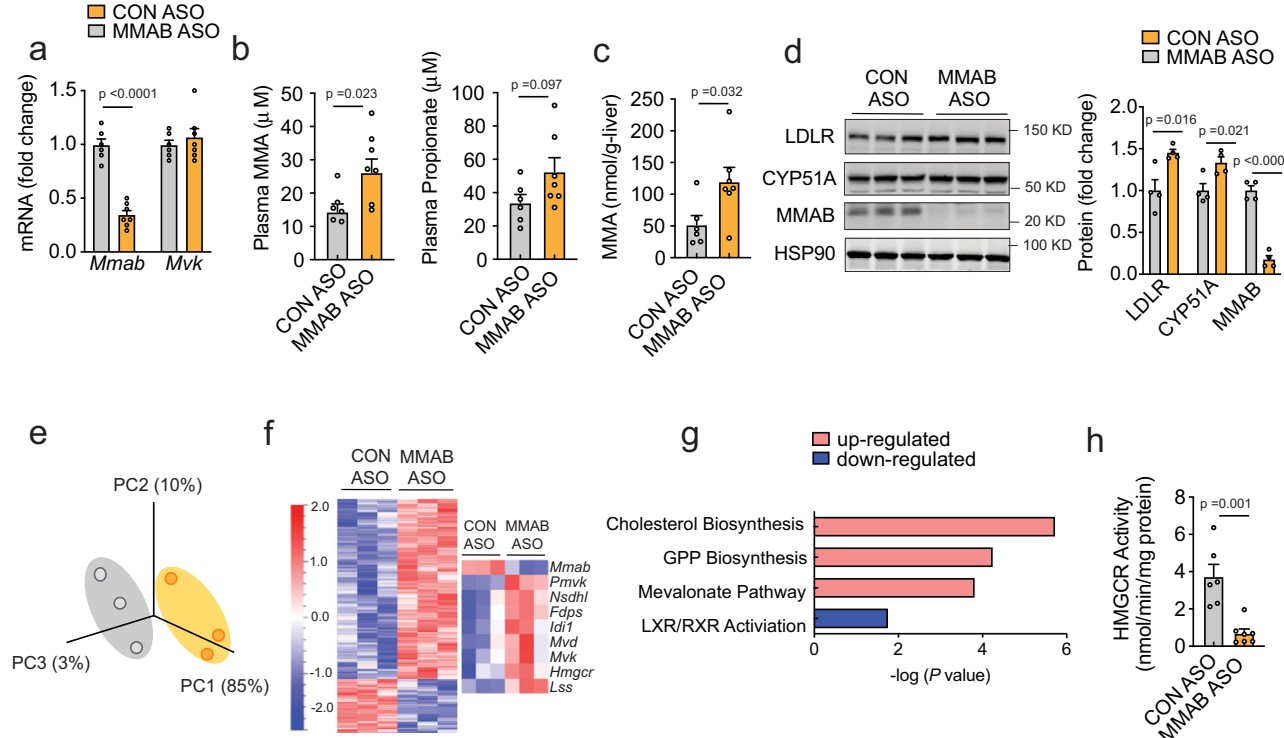

**Fig. 7 Knockdown of MMAB reduces HMGCR activity and increases SREBP2-mediated gene expression in vivo. a** qRT-PCR expression of *Mmab* and *Mvk* in 8-week-old C57BL/6 (WT) mice treated weekly with 50 mg/kg MMAB antisense oligonucleotide (MMAB ASO) or 50 mg/kg control antisense oligonucleotide (CON ASO) for 6 weeks. Data are mean ± s.e.m. *n* = 6 CON ASO, 7 MMAB ASO. Statistical comparisons by unpaired two-sided Student's *t* test. **b** Plasma methylmalonic acid (MMA) and propionate in mice treated as in (**a**). Data are mean ± s.e.m. *n* = 6 CON ASO, 7 MMAB ASO. Statistical comparisons by unpaired two-sided Student's *t* test. **c** Hepatic methylmalonic acid (MMA) in mice treated as in (**a**). Data are mean ± s.e.m. *n* = 6 CON ASO, 7 MMAB ASO. Statistical comparisons by unpaired two-sided Student's *t* test. **d** Representative Western blot analysis of MMAB and SREBP2-responsive genes in the livers of mice treated as in (**a**). HSP90 was used a loading control. Quantification of blot shown to the right. Data are mean ± s.e.m. *n* = 4 mice per group. Statistical comparisons by unpaired two-sided Student's *t* test. **e** Principle component (PC) analysis of RNA-seq data from liver tissue of mice treated as in (**a**). *n* = 3 mice per group. **f** Heatmap analysis of hepatic genes differentially expressed between CON ASO and MMAB ASO treated mice. Data are represented as Log2 (MMAB ASO/CON ASO treated) values. Genes in blue are downregulated, while genes in red are upregulated ($P \leq 0.05$; FC $\geq$ 1.5, FC $\leq$ −1.5). Differentially expressed cholesterol biosynthetic genes are shown to the right (FDR adjusted, $P \leq 0.05$; FC $\geq$ 1.5, FC $\leq$ −1.5). *n* = 3 mice per group. **g** Pathway enrichment analysis of differentially expressed genes in the livers of CON ASO and MMAB ASO treated mice. Pathways with increased activity (positive *Z*-score) are highlighted in red, while pathways with decreased activity (negative *Z*-score) are highlighted in blue. See Supplementary Data 6. **h** HMGCR activity in the livers of mice treated as in (**a**). Data are mean ± s.e.m. *n* = 6 CON ASO, 7 MMAB ASO. Statistical comparisons by unpaired, two-sided Mann−Whitney test. Source data are provided as a Source Data file.

identified in our genomic screen remain to be determined, the likelihood that they represent regulators of LDLR activity is high.

From the candidates validated in our functional genomic screens, we initially focused on MMAB and provide multiple lines of evidence for its role in regulating LDLR activity and cholesterol homeostasis. Intriguingly MMAB is located in a head-to-head conformation with MVK and contains several conserved NFY and SREBP2-binding sites[24]. Consistent with previous studies, here, we demonstrate that MMAB expression is modulated by intracellular cholesterol levels in vitro and in vivo. Moreover, we also demonstrate that MMAB promoter activity is activated by SREBP2, similarly to MVK, thereby establishing MMAB as a valid SREBP2-target gene. Intriguingly, genome-wide profiling in Huh7 cells also revealed that other members of the vitamin B$_{12}$ pathway were modulated by intracellular cholesterol levels (*TCN1*, *LMBRD1*, *MUT*, and *MMAA*; Supplementary Data 2), suggesting that the cobalamin and cholesterol biosynthetic pathway are tightly linked.

Our high-throughput RNAi screen demonstrated that loss of MMAB leads to increased LDLR expression and activity. Importantly, these findings were validated using several different RNAi technologies (pooled RNAi screen, deconvolution screen,

independent siRNA library, antisense oligonucleotides, and GapMers), in multiple cell lines (human and mouse hepatocytes), and in vivo (C57BL/6 and *Apobec1*$^{−/−}$; *Ldlr*$^{+/−}$ mice). Mechanistically, siMMAB-mediated increases in LDLR activity were due to reductions in cholesterol biosynthesis and subsequent upregulation of SREBP2-mediated gene expression. While knockdown of MMAB caused a relatively small reduction in intracellular cholesterol levels (~20%), these results are consistent with cholesterol homeostasis being tightly regulated. Indeed, a 5% depletion of ER-cholesterol content is sufficient to induce SREBP2 transport from the ER to the Golgi, SREBP2 cleavage and transcriptional activation of SREBP2-responsive genes (LDLR)[43].

How could a gene involved in vitamin B$_{12}$ synthesis alter cholesterol biosynthesis and SREBP2-mediated gene expression? Loss of MMAB function leads to a reduction in adenosylcobalamin synthesis, a cofactor for the mitochondrial enzyme, MUT, which catalyzes the isomerization of methylmalonyl-CoA to succinyl-CoA. Deficiencies in *MUT* or *MMAB* lead to elevated levels of propionic acid and methylmalonic acid in the inherited metabolic disorder methylmalonic aciduria[29]. Intriguingly, we were able to phenocopy the effects of MMAB knockdown on cholesterol metabolism with propionic acid and methylmalonic

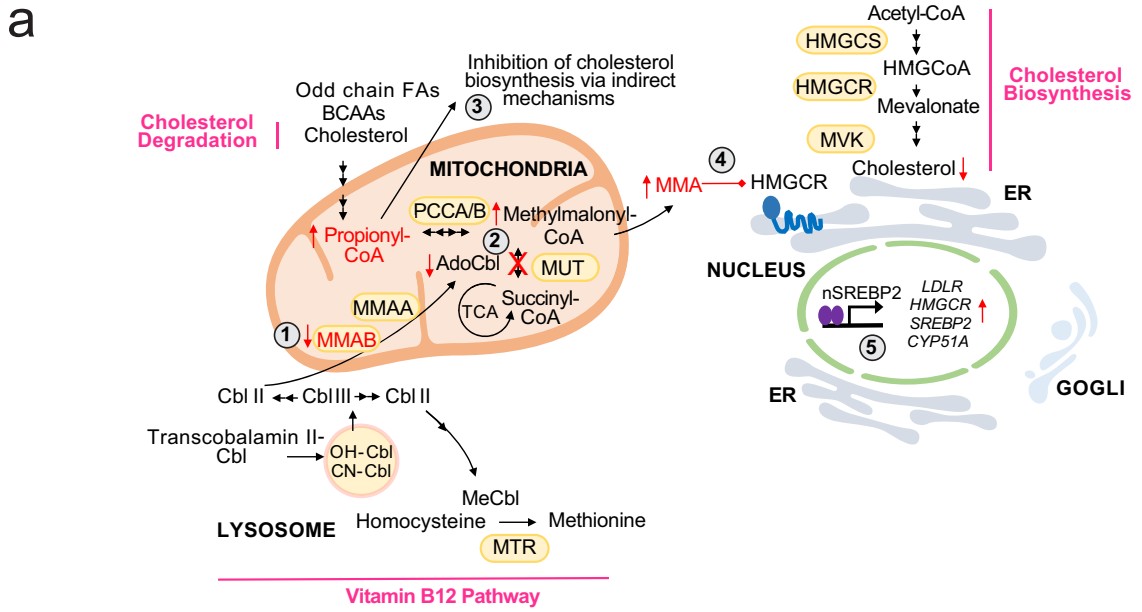

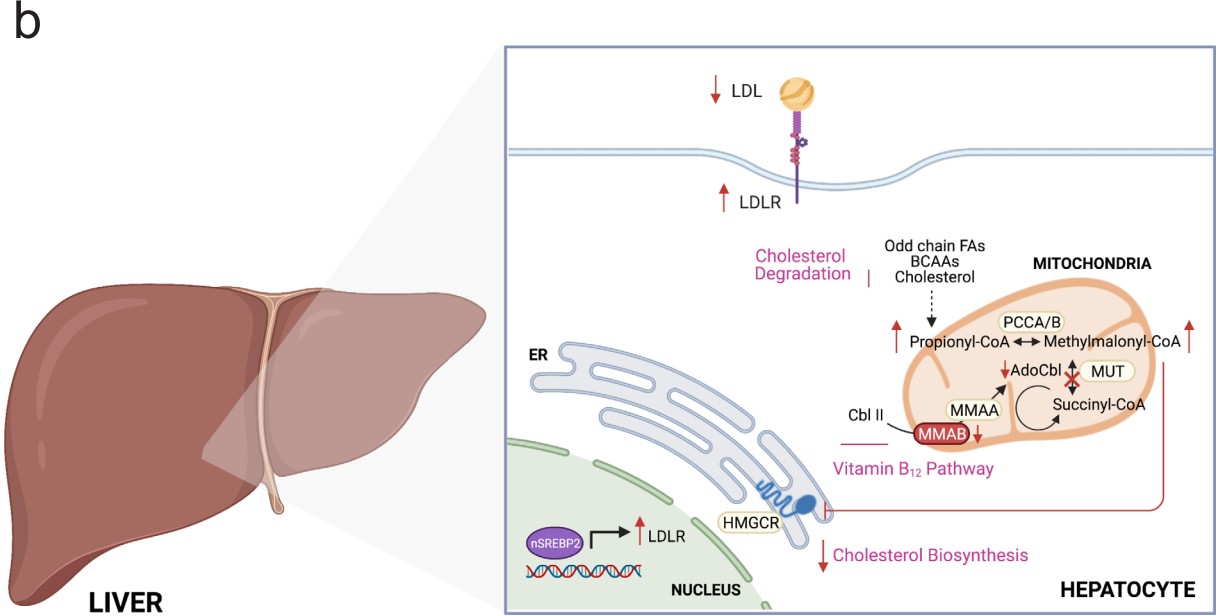

**Fig. 8 Feedback control of cholesterol biosynthesis through modulation of adenosylcobalamin metabolism. a** Schematic showing negative feedback regulation of cholesterol biosynthesis by MMAB. Loss of MMAB leads to reduced conversion of cob(I)alamin to AdoCbl (**1**), which impairs the function of methylmalonyl-CoA mutase (MUT) activity and results in the incomplete breakdown of certain amino acids, cholesterol and odd-chain fatty acids (**2**). This defect leads to the accumulation of propionic acid, which reduces cholesterol biosynthesis through indirect mechanisms (**3**), and the accumulation of methylmalonic acid, which reduces cholesterol biosynthesis by inhibiting HMGCR (**4**). Sterol depletion leads to the subsequent upregulation of SREBP2-mediated gene expression (**5**), thereby providing an additional control point by which cholesterol biosynthesis is regulated by its end product. BCAAs branched chain amino acids, FAs fatty acids, MMA methylmalonic acid. **b** Schematic showing the role of the adenosylcobalamin pathway in regulating hepatic LDLR expression and circulating LDL-C levels.

acid in vitro—e.g. propionic and methylmalonic acid increased SREBP2-mediated gene expression and LDLR activity in Huh7 cells. In support of this, intracellular cholesterol levels and LDLR expression and activity were also significantly altered in primary human fibroblasts from patients with methylmalonic aciduria. Interestingly, the mitochondrial dysfunction caused by the lack of *MUT* activity can result in the accumulation of hepatic neutral lipids and mild/moderate hypertriglyceridemia[44,45]. Moreover,

some patients with methylmalonic aciduria have been reported to have low post-prandial circulating HDL-C levels[44], in support of previous GWAS studies identifying *MMAB* as a susceptibility gene influencing HDL-C levels in humans[27].

While the role for propionate in regulating cholesterol homeostasis in vivo is mixed[46–50], our results are consistent with reports demonstrating a role for propionate in regulating hepatic cholesterol biosynthesis[49,50] and plasma levels of total cholesterol

in rodents[46,49,50] and humans[51]. In particular, Hara et al.[50] demonstrated that rates of hepatic cholesterol biosynthesis were decreased in rats fed a diet rich in SCFAs compared to fiber-free-fed controls. Interestingly, plasma levels of total cholesterol were also significantly reduced and negatively correlated with levels of portal plasma propionate (~250 µM) 5 h post-SCFA-feeding[50]. While the role of acetate and butyrate in mediating SCFA-mediated reductions in hepatic cholesterol biosynthesis cannot be ruled out, our findings suggest that the SCFA-mediated reductions in total plasma cholesterol may be due to SREBP2-mediated upregulation of hepatic LDLR.

The mechanism by which propionate reduces cholesterol biosynthesis remains to be determined, Initially, we hypothesized that propionic acid may reduce cholesterol biosynthesis by inhibiting HMGCR, the rate-limiting enzyme of cholesterol biosynthesis. Increasing doses of propionate (10 µM−100 mM), however, did not alter HMGCR activity in isolated murine hepatic microsomes, despite reducing $^{14}$C-acetate incorporation into cholesterol at physiologic concentrations (100 µM)[32,33,46] in human hepatic (Huh7) cells. As such, inherent differences may exist in the regulation of cholesterol biosynthesis between primary human and rodent hepatocytes and/or cultured cell lines. Indeed, supraphysiologic concentrations (10−20 mM) of propionate are needed for 50% inhibition of cholesterol biosynthesis in primary human hepatocytes, in contrast to primary rat hepatocytes which achieve 50% inhibition with 100 µM propionate[30]. Alternatively, it is possible that propionate's cholesterol-lowering effects are due to AMPK-mediated inhibition of HMGCR, given the effect of short-chain fatty acids on AMPK activation[52].

We next sought to determine how methylmalonic acid affects LDLR expression and activity. Intriguingly, we found that methylmalonic acid was able to significantly reduce HMGCR activity by ~50% at concentrations as low as 50 µM. These findings are consistent with our in vivo studies (which demonstrate that loss of MMAB in mice results in a marked increase in plasma and hepatic methylmalonic acid levels and subsequent reduction in hepatic HMGCR activity and sterol content), and in line with other studies suggesting the importance of methylmalonic acid in regulating lipid metabolism. For example, in rat cerebral cortex and liver, 5 mM MMA was recently shown to reduce $^{14}$C-acetate incorporation into total lipids[53,54]. Moreover, chronic administration of methylmalonic acid (plasma concentrations of ~2 mM) to Wistar rats markedly reduced plasma triglyceride levels and cerebral myelin content. While these results highlight the underappreciated role of methylmalonic acid in regulating lipid metabolism, no changes were observed in circulating plasma cholesterol levels or total cholesterol content in the rat cerebellum. Given the differences in lipoprotein metabolism between rodents and humans, further studies are warranted to dissect the effects of methylmalonic acid on hepatic cholesterol biosynthesis, LDLR expression and circulating LDL-C levels in vivo. Indeed, we were only able to detect differences in plasma LDL-C when we inhibited MMAB expression in Western-diet fed $Apobec1^{-/-};Ldlr^{-/+}$ mice.

Collectively, this study highlights a previously unidentified link between AdoCbl and cholesterol metabolism (Fig. 8a, b). In mammals, vitamin B$_{12}$ is acquired through dietary uptake and is processed into its cofactor forms, methylcobalamin (MeCbl) and AdoClb. Interestingly, MeCbl (a cofactor for 5-methyltetrahydrofilic acid (MTHF)-dependent methionine synthase) has previously been linked to the regulation of SREBP stability through decreased S-adenosylmethionine (SAMe) and subsequent reductions in phosphatidylcholine (PC)[55]. Specifically, Walker et al.[55] identified a conserved negative feedback circuit whereby SREBP1 controls genes in the one-carbon cycle to produce SAMe, maintain PC levels and affect its processing and nuclear accumulation. Here, we describe a separate feedback loop whereby SREBP2 regulates MMAB expression to control AdoCbl production, intracellular propionate and methylmalonic acid levels, SREBP2-mediated gene expression and LDLR activity (Fig. 8). In line with this, other genes affecting AdoCbl levels were also identified in our genome-wide screen, including CLYBL (Supplementary Data 1), a ubiquitously expressed mitochondrial enzyme that operates as a citramalyl-CoA lyase in mammalian cells[56]. Shen et al. recently demonstrated that loss of CLYBL leads to a defect in MUT activity through itaconyl-CoA-mediated reductions in AdoCbl. As CLYBL deficiency leads to increased levels of propionic acid and methylmalonic acid[56], it is plausible that decreased levels of CLYBL led to increased LDLR activity in our screen due to altered levels of propionate and/or methylmalonic acid and SREBP2-mediated increases in LDLR expression.

In conclusion, our studies unequivocally establish a role for MMAB in regulating LDLR activity and cholesterol metabolism, providing an additional control point by which cholesterol biosynthesis is regulated by its end product. Importantly, human genetic data suggest that MMAB has a prominent role in governing cholesterol metabolism in humans, as SNPs in the MVK/MMAB locus are strongly associated with altered levels of HDL-C, LDL-C, and CVD risk. Here we provide insight into the mechanism by which these variants might alter LDL-C in humans: by reducing MMAB activity, increasing methylmalonic acid accumulation, inhibiting HMGCR activity and cholesterol biosynthesis, increasing SREBP2-mediated gene expression, and consequently LDLR expression and activity (Fig. 8b). Further experiments are therefore warranted to dissect the contribution of these variants to altered lipid levels and CVD risk.

## Methods

**Cell culture.** Age- and sex-matched fibroblasts from patients with methylmalonic aciduria due to MUT deficiency (MUT fibroblasts, #GM00050) and otherwise healthy individuals (WT fibroblasts, #GM05659) were obtained from the NIGMS Human Genetic Cell Repository (Coriell Institute for Medical Research). According to the DHHS Office for Human Research Protections (OHRP) Guidance on Research Involving Coded Private Information or Biological Specimens, biospecimens obtained from NIGMS are not considered to be human subjects because the research conducted with these samples does not involve intervention or interaction with the individuals and all samples do not contain identifiable private information (45 CFR 46.102(f)). Therefore, experiments conducted with these fibroblasts did not require approval from the Yale HRPP (HRPP Policy 440.1).

The human hepatocellular carcinoma cell line, Huh7 (JCRB Cell Bank JCRB0403), and mouse hepatic cell line (Hepa1–6; ATCC CRL-1830) were a kind gift from Dr. Edward Fisher (NYU School of Medicine). HeLa cells were purchased from ATCC (ATCC CCL-2). Huh7, Hepa, and HeLa cells were maintained in Dulbecco's modified Eagle medium DMEM) (Corning #10-013-CV) containing 10% (v/v) FBS (Sigma #12306C) and 2% (v/v) penicillin-streptomycin (Gibco #15140122) in 100 mm dishes at 37 °C and 5% CO$_2$. Cells were split every 3 days. Before performing experiments all cell lines were authenticated by morphology check at high and low plating densities, growth curve analyses, and periodic assessment of mycoplasma infection. WT and MUT primary human fibroblasts were maintained in DMEM containing 20% (v/v) FBS and 2% (v/v) penicillin-streptomycin in 100 mm dishes at 37 °C and 5% CO$_2$ prior to experimentation; all studies were performed within 2–3 passages. For DiI-LDL uptake/binding experiments, gene expression analysis and sterol/metabolite measurements, cells were cultured in DMEM containing 10% LPDS as described below. LPDS was prepared from FBS delipidated with 4% fumed silica (Sigma #S5505)[57]. Human LDL was isolated and labeled with the fluorescent probe DiI as previously reported[58]. 1,1′-Dioctadecyl-3,3,3,3′-tetramethylindocarbocyanineperclorate (DiI) was purchased from Sigma (#468495).

**RNAi screen.** All steps of the genome-wide screen, including reverse transfection and image acquisition and analysis, were performed at the NYU RNAi Core Facility (NYU School of Medicine).

*Reverse transfection, fixation and staining.* Huh7 cells were reverse transfected in triplicate with a total of 65,058 siRNAs targeting 21,686 genes (Ambion$^{TM}$ Silencer Human Genome siRNA library V3) in Corning 384-well flat clear-bottom black plates (Fisher Scientific) using a standard reverse transfection protocol as we

previously described[21]. Briefly, Huh7 cells (5000 cells/well in 30 μl of DMEM media containing 10% LPDS) were seeded into a well containing 30 μl of transfection mix (25 μl of Opti-MEM™ medium (ThermoFisher #31985070), 0.07 μl RNAi Max (Invitrogen #13778150), and 5 μl of 0.3 μM siRNA). 20 μl of fresh LPDS media was added to all wells 12 h post transfection, giving a final siRNA concentration of 18 nM. Forty-eight hours later, cells were incubated with 10 μl of fresh LPDS containing 30 μg of DiI-LDL-cholesterol/ml for 8 h at 37 °C. Following incubation, cells were washed twice with 1× PBS and fixed with 4% PFA (w/v) pH 7.4 in 1× PBS for 15 min. After three subsequent washes with 1× PBS (Gibco #14190-144), cells were incubated with PBS containing 1 μg/ml Hoechst (Invitrogen #H3570) for 25 min. Before scanning, a final wash with 1× PBS was performed and plates were spun down to minimize contaminants when imaging with the automated microscope. All liquid handling steps, including seeding, DiI-LDL incubation, fixation, washing, and Hoechst incubation, were performed using a Wellmate Microplate Dispenser (Matrix Technologies) and BioTek Plate Washer (PerkinElmer). The triplicate screen consisted of 15 384-well plates and was completed over the course of 4 days.

*Image acquisition and analysis.* Automated high content and throughput images were acquired using an Arrayscan VTI HCS Reader (Thermo Scientific) with a Zeiss ×10 objective as we previously described[21]. 384-well plates were loaded onto the microscope using a Catalyst Express robotic arm and imaged overnight. In each well, cell nuclei and DiI-LDL intensities were imaged in five pre-defined fields. Image data were analyzed using BioApplication's Target Activation V3 image analysis software (Thermo Scientific). Briefly, nuclei were first identified on the Hoechst stain (Channel 1). Following this, cell boundaries were estimated using the geometric segmentation method and used to calculate DiI intensity (Channel 2) within each cell. In total, valid object count, mean average intensity, and total average intensity of DiI were recorded for each field.

*Hit classification.* siRNAs were scored based on their ability to significantly increase or decrease DiI intensity compared to negative controls. Cytotoxic siRNA over-expression phenotypes were filtered for hit classification by excluding wells in which fewer than 500 cells were identified as valid objects. In addition, validated internal controls, including non-silencing (NS) siRNA (Dharmacon siGENOME non-targeting control pool #2, #D-001206-14-20), siLDLR (Dharmacon siGENOME SMARTPool, #M-011073-01-0020), and siNPC1 (Dharmacon siGENOME SMARTPool, #M-008047-01-0020), as well as the negative control siRNAs and siRNA KIF11 (Life Technologies), were used on each plate to monitor transfection efficiency. Mean average intensities of each well were normalized to plate medians and converted to robust z scores using MATLAB, as previously described[59]. Robust z scores were compared between each plate replicate and the mean of each score was calculated and used to rank potential candidates. Those siRNAs that had a robust z score of ≤−2.0 and ≥2.0 (1491 hits) were chosen for further characterization. To narrow down candidate siRNA genes for the deconvolution screen (see below), hits were subjected to several screening passes (Fig. 1a). Briefly, candidates were chosen based on whether their expression was significantly modulated by intracellular cholesterol levels (Affymetrix arrays) and/or whether they had SNPs that were previously linked to alterations in plasma cholesterol/CVD risk (250 hits).

**Deconvolution screen.** To validate candidates from the primary RNAi screen, 250 candidate genes were subjected to a second microscope-based screening assay with three individual siRNAs from the initial siRNA pool used in the primary RNAi screen (Ambion™ *Silencer* Human Genome siRNA library). After confirming efficient transfection efficiency, mean average intensities of each well were normalized to plate medians and converted to robust z scores using MATLAB, as previously described[59]. Robust z scores were compared between each plate replicate and the mean of each score was calculated and used to rank potential candidates. Those siRNAs that produced a similar phenotypic effect as in the primary RNAi screen and had a robust z score of ≤−1.6 and ≥1.6 were considered validated "hits" (160 of the original 250 genes analyzed, 64%).

**Secondary screening assay with Dharmacon library.** To further validate candidates from the primary and deconvolution screen, a subset of candidate genes was subjected to a third microscope-based screening assay with a pool of siRNAs from Dharmacon (ON-TargetPlus SMARTPools). siRNAs were spotted onto 384-well plates and screened in an identical format as the primary and deconvolution screen. A non-silencing (NS) siRNA (ON-TargetPlus non-targeting pool #D-001810-10-20), siLDLR (ON-TargetPlus SMARTPool L-011073-00-005), and siNPC1 (ON-TargetPlus SMARTPool #L-008047-00-0005) were used as negative and positive controls for the assay. Mean average intensities of each well were normalized to plate medians and converted to robust z scores using MATLAB as described above. Robust z scores were compared between each replicate and the mean of each score was calculated and used to rank potential candidates. Those siRNAs that produced a similar phenotypic effect (robust z score of ≤−1.6 and ≥1.6) as in the primary RNAi screen were selected for further analysis.

**Affymetrix arrays.** Huh7 cells were cultured in DMEM containing 10% FBS and 120 μg/ml nLDL (to cholesterol enrich), 5 μM statin (to cholesterol deplete, Sigma #S6196) or vehicle control for 24 h. Following this, cells were harvested in Trizol (Invitrogen #15596026) and RNA was purified using the RNeasy Isolation Kit (Qiagen #74104). The purity and integrity of total RNA was verified using the Agilent Bioanalyzer (Agilent Technologies, Santa Clara, CA) and hybridized to the Affymetrix GeneChip Human Genome U133 Plus 2.0 Array according to the manufacturer's instructions. Two biological replicates for each condition were used for microarray analysis. Raw data were normalized and analyzed by GeneSpring GX software version 11.5 (Agilent Technologies). mRNAs showing altered expression with nLDL/statin treatment compared to control were identified using unpaired two-sided Student's t test ($P ≤ 0.05$, FC ≥ 1.1 or ≤−1.1). Functional annotation clustering of up- and downregulated genes was performed using DAVID (see below).

**Bioinformatics analysis of top candidates from primary screens.** Functional annotation clustering and enrichment scoring of the 7990 genes from the Affymetrix arrays were performed using DAVID v6.8 (http://david.abcc.ncifcrf.gov). "High" classification stringency settings yielded 153 functional annotation clusters ($P ≤ 0.05$) (Supplementary Data 2). In another set of analyses, we took the top 250 candidate genes identified in the primary RNAi screen and Affymetrix arrays and uploaded them into the Molecular Signature Database (http://software.broadinstitute.org) to find enriched GO terms (FDR q value = 0.001). Functional annotation clustering of these 250 genes was then performed using DAVID and used to identify functional interactions between candidate genes using STRING v9.05 (http://string-db.org). Enrichment maps of DAVID and STRING outputs were made using MATLAB, the Cytoscape plugin (v3.4) and EnrichmentMap plugin (v3.0) as previously described[60,61]. Smaller annotation clusters and unconnected genes were left out of some visualization networks due to space constraints.

**siRNA transfections.** For siRNA transfections, Huh7 cells were plated in six-well plates or 100 mm dishes and transfected with 20–60 nM of siRNAs against MMAB (ON-TargetPlus SMARTPool #L-015137-00-0005), MUT (ON-TargetPlus SMARTPool #L-008429-00-0005), MVK (siGENOME SMARTPool #M-006749-00-0005), and corresponding negative control siRNAs (ON-TARGETPlus non-targeting pool #D-001810-10-20 or siGENOME non-targeting control pool #D-001206-14-20) for 48–60 h in Opti-MEM™ medium using RNAimax (Invitrogen) according to the manufacturer's instructions as previously described[62]. Following 12 h of transfection, DMEM media containing 10% LPDS and 2% penicillin-streptomycin was added and cells were incubated at 37 °C and 5% $CO_2$ for an additional 12 h. For those cells simultaneously transfected with an siRNA against MMAB (40 nM) and MUT (40 nM), cells were compensated with an equal amount of NS siRNA (40 nM) in the single-transfection groups to control for off-target effects (total siRNA concentration per well, 80 nM). In another set of experiments, Hepa cells were transfected with 50 nM of four different antisense locked nucleic acid (LNA)™ GapmeRs against MMAB (Exiqon #300600) or scrambled control (Exiqon #300610) for 72 h. Verification of knockdown was assessed by western blotting and/or qRT-PCR analysis as described below.

**MUT fibroblasts.** WT and *MUT* fibroblasts were plated in six-well plates and cultured in DMEM supplemented with 10% (v/v) LPDS for 24 h. Following incubation, cells were washed once in 1× PBS, RNA and protein extracted and gene expression assessed as described below. In another set of experiments, WT and *MUT* fibroblasts were cultured in six-well plates with 10% (v/v) LPDS for 24 h and incubated with 30 μg/ml DiI-LDL for 2 h at 37 °C; DiI-LDL uptake was assessed by immunofluorescence and FACS as described below. In another set of experiments, WT and *MUT* fibroblasts were plated in 100 mm dishes and cultured in DMEM supplemented with 10% (v/v) LPDS for 24 h; sterol content and relative levels of metabolites were measured as outlined below.

**Propionic and methylmalonic acid studies.** For propionic acid studies, Huh7 cells were cultured in DMEM supplemented with 10% (v/v) LPDS for 24 h and incubated in culture media containing 5 μM statin, the indicated doses of propionic acid sodium salt (Sigma #P5436) or vehicle control (ethanol) for an additional 24 h. For methylmalonic acid studies, Huh7 cells were cultured in Opti-MEM™ with RNAiMAX (ThermoFisher) for 24 h to enhance uptake of methylmalonic acid. After this pretreatment, cells were incubated in DMEM containing 10% (v/v) LPDS containing the indicated doses of methylmalonic acid (Sigma #M54058), 5 μM statin or vehicle control (ethanol) for an additional 24 h. Following treatment, gene expression, LDLR activity, total sterol content, and/or $^{14}C$-acetate incorporation into cholesterol were assessed as described. In another set of experiments, cells were pretreated with 10 μM S1P inhibitor (PF-429242; Sigma #SML0667) for 2 h prior to addition of statin or propionate; gene expression was assessed as described below.

**LDL receptor activity assays.** In one set of experiments, Huh7 cells were transfected in six-well plates with 40 nM of an siRNA against MMAB or NS control siRNA in DMEM containing 10% (v/v) LPDS for 48 h as described above. In another set of experiments, Huh7 cells were incubated in 10% (v/v) LPDS for 24 h

and then treated with the indicated doses of statin, propionic acid, methylmalonic acid or ethanol (vehicle control) for an additional 24 h. Following treatment, cells were washed once in 1× PBS and incubated in fresh media containing DiI-LDL (30 μg cholesterol/ml) for 2–8 h. Non-specific uptake was determined in extra wells containing a 50-fold excess of unlabeled native LDL (nLDL). For DiI-LDL uptake experiments, cells were incubated at 37 °C to allow for endocytosis and recycling of the LDLR (incubated for 8 h in screening optimization experiments and for 2 h at 37 °C for subsequent validation experiments)[57,58]. For DiI-LDL binding experiments, cells were washed once in ice-cold 1× PBS and incubated with DiI-LDL (30 μg cholesterol/ml) for 90 min at 4 °C to inhibit endocytosis and internalization of the LDLR[57,58]. At the end of the incubation period, cells were washed once in 1× PBS, resuspended in 1 ml of PBS and analyzed by flow cytometry (FACScalibur, Becton Dickinson) as previously described[63]. The results are expressed in terms of specific median intensity of fluorescence (M.I.F.) after subtracting autofluorescence of cells incubated in the absence of DiI-LDL. Representative gating strategies are provided in Supplementary Fig. 9.

**Fluorescence microscopy.** For DiI-LDL uptake assays, Huh7 cells were grown on coverslips and transfected with 40 nM of an siRNA against MMAB or NS control siRNA in DMEM containing 10% (v/v) LPDS. Forty-eight hours post transfection, cells were incubated with DiI-LDL (30 μg/ml cholesterol) for 2 h at 37 °C as previously described[21]. Then, cells were washed twice with 1× PBS, fixed with 4% (w/v) PFA, and blocked (3% (w/v) BSA in 1× PBS) for 15 min. Following this, cells were washed twice and mounted on glass slides with Prolong-Gold (Cell Signaling Technologies #8961). All images were analyzed using confocal microscopy (Leica SP5 II) equipped with a ×63 Plan Apo Lenses. All gains for the acquisition of comparable images were maintained constant. Analysis of different images was performed using ImageJ (NIH) and Adobe Photoshop CS5. In another set of experiments, WT and *MUT* fibroblasts were grown in six-well plates containing DMEM with 10% (v/v) LPDS for 24 h. Following this, cells incubated with DiI-LDL (30 μg/ml cholesterol) for 2 h at 37 °C and imaged using an EVOS digital inverted florescence microscope (AMG).

**RNA isolation and quantitative real-time PCR.** Total RNA was isolated using TRIzol reagent (Invitrogen) and purified using the RNeasy isolation kit (Qiagen) according to the manufacturer's protocol. 1 μg of total RNA was reverse transcribed using iScript RT Supermix (Bio-Rad #1708841), following the manufacturer's protocol. Quantitative real-time PCR (qRT-PCR) analysis was performed in duplicate using iQ SYBR green Supermix (Bio-Rad #1708880) on an iCycler Real-Time Detection System (Bio-Rad). The mRNA level was normalized to GAPDH or 18S as a housekeeping gene. For mouse tissues, total liver RNA from C57BL6 and *Apobec1*−/−;*Ldlr*−/+ mice was isolated using the Bullet Blender Tissue Homogenizer (Next Advance) in TRIzol and purified using the RNeasy isolation kit (Qiagen). 1 μg of total RNA was reverse transcribed and gene expression assessed as above. Primer sequences are listed in Supplementary Table 1.

**RNA-seq.** Total RNA from liver samples of mice treated with 50 mg/kg control ASO or MMAB ASO was extracted and purified using the RNeasy isolation Kit (Qiagen) according to the manufacturer's instructions and treated with DNAse to remove genomic contamination (RNA MinElute Cleanup, Qiagen). With the technical support of the Yale Center for Genome Analysis, the purity and integrity of total RNA was verified using the Agilent Bioanalyzer (Agilent Technologies, Santa Clara, CA). rRNA was depleted from the RNA samples using Ribo-Zero rRNA Removal Kit (Illumina). RNA libraries were made using TrueSeq Small RNA Library preparation (Illumina) and were sequenced for 45 cycles on Illumina HiSeq 2000 platform (1 × 100 bp read length). Using the PartekFlow® software, version 8.0.19.0405 (Partek, Inc., St. Louis, MO), the reads were aligned with the STAR algorithm to the mouse genome *mmu10*. The quantify to annotation model (Partek E/M) algorithm was used to estimate the transcript expression abundance. Counts were normalized using the recommended methods (Total count, Add 0.0001, Log2). To identify differential expression patterns, the Differential Gene Expression-GSA algorithm was implemented. A default *P* value ≤ 0.05 was considered statistically significant with a fold-change ≥ 1.5 for upregulated transcripts or ≤−1.5 for downregulated transcripts. Heat maps and principal component (PC) analysis plots were generated using Qlucore Omics Explorer v 3.2 (Qlucore AB, Lund, Sweden). Ingenuity Pathway Analysis Spring Release 2019 (Ingenuity Systems QIAGEN, Redwood City, CA, USA) was used to carry out pathway enrichment analyses for differentially expressed genes across samples.

**Western blot analysis.** Unless otherwise stated, cells (confluent well of a six-well plate) and liver tissue (~50 mg) were lysed in ice-cold VJ lysis buffer containing 50 mM Tris-HCl, pH 7.5, 125 mM NaCl, 1% (v/v) NP-40, 5.3 mM NaF, 1.5 mM NaP, 1 mM sodium orthovanadate, and 1 mg/ml of protease inhibitor cocktail (Roche #04693159001) and 0.25 mg/ml AEBSF (Roche #11427393103). Cell lysates were rotated at 4 °C for 30 min before the insoluble material was removed by centrifugation at 12,000 × *g* for 10 min. After normalizing for equal protein concentration, cell lysates were resuspended in reducing loading sample buffer, boiled for 5 min and separated by SDS-PAGE. Following overnight transfer of the proteins onto nitrocellulose or PVDF membranes, the membranes were blocked with

5% BSA (w/v) in wash buffer and probed with the following antibodies overnight at 4 °C: LDLR (Abcam #ab30532, 1:1000), MMAB (Novus #NBP1-86602, 1:1000), NPC1 (Novus #NB400-148, 1:500), HSP90 (BD Biosciences #610418, 1:1000), CYP51A (Proteintech #13431-1-AP, 1:500), ACTIN (Abcam #ab8227, 1:1000), and DHCR24 (Cell Signaling #2033, 1:1000).

For the detection of SREBP2, adherent cells were harvested with trypsin-EDTA and centrifuged at 300 × *g* for 5 min at 4 °C. Cells were then washed one time with ice-cold PBS, snap frozen in liquid nitrogen and stored at −80 °C until time of lysis. Following the addition of ice-cold SREBP lysis buffer containing 20 mM Tris-HCl pH 8, 120 mM KCl, 1 mM DTT, 1 mM EDTA, 2 mM EGTA, 0.1% Triton-X, 0.5% NP-40, 1 mM benzamidine HCl, 10 μg/ml Antipain, 1 μg/ml Leupeptin, 40 μg/ml Aprotinin, 100 mM NaF, 20 mM Na$_3$MoO$_4$, 20 mM β-glycerophosphate, 2 mM Na$_3$VO$_4$, 1 mM PMSF (Sigma #329-98-6), and 1 μg/ml caspase-3 inhibitor Ac-DMQD-CHO (Cayman Chemical #27103), samples were vortexed, briefly sonicated and passed through a 22G needle ten times. Lysates were then incubated on ice for 30–60 min (vortexing every 10 min) before the insoluble material was removed by centrifugation at 12,000 × *g* for 10 min at 4 °C. After normalizing for equal protein concentration, cell lysates were resuspended in reducing loading sample buffer, boiled for 5 min and separated by SDS-PAGE. Following overnight transfer of the proteins onto nitrocellulose or PVDF membranes, the membranes were blocked with 5% BSA (w/v) in wash buffer and probed with the following antibodies overnight at 4 °C: SREBP2 (BD Biosciences #557037, 1:500).

HMGCR was detected as previously described[64] with the following modifications. Briefly, confluent cells of a six-well plate were harvested with trypsin-EDTA and centrifuged at 300 × *g* for 5 min at 4 °C. Cell pellets were then washed one time with ice-cold 1× PBS, snap frozen in liquid nitrogen and stored at −80 °C until time of lysis. Cell pellets were lysed in Buffer A containing 10 mM Tris-HCl pH 6.8, 1% (w/v) SDS, 100 mM NaCl, 1 mM EDTA and 1 mM EGTA with protease inhibitors. Lysates were incubated on ice for 30–60 min (vortexing every 10 min) before the insoluble material was removed by centrifugation at 12,000 × *g* for 10 min at 4 °C. After normalizing for equal protein concentration, lysates were mixed in a 1:1 ratio with Buffer B (62.5 mM Tris-Cl pH 6.8, 15% (w/v) SDS, 8 M Urea, 15% glycerol and 100 mM DTT) and reducing loading sample buffer, heated at 37 °C for 20 min and separated by SDS-PAGE. Following overnight transfer of the proteins onto nitrocellulose or PVDF membranes, the membranes were blocked with 5% non-fat milk (w/v) in wash buffer and probed with the following antibodies overnight at 4 °C: HMGCR (culture supernatant of the mouse hybridoma, A9 cell line [ATCC #CRL-1811], 1:10).

For detection of LDLR with an antibody from Cayman Chemical, cells were lysed in ice-cold LDLR lysis buffer containing 50 mM Tris-HCl (pH 6.8), 2.5 mM CaCl$_2$, 1% (v/v) Triton-X 100, 5.3 mM NaF, 1.5 mM NaP, 1 mM sodium orthovanadate and 1 mg/ml of protease inhibitor cocktail (Roche). Cell lysates were rotated at 4 °C for 30 min before the insoluble material was removed by centrifugation at 12,000 × *g* for 10 min. After normalizing for equal protein concentration, cell lysates were resuspended in non-reducing sample buffer and separated by SDS-PAGE. Following overnight transfer of the proteins onto nitrocellulose membranes or PVDF, the membranes were blocked with 5% BSA (w/v) in wash buffer and probed with the following antibodies overnight at 4 °C: LDLR (Cayman Chemical #10007665; 1:1000). For detection of LDLR in tissues, ~50 mg of liver was homogenized in ice-cold VJ lysis buffer as described above. After normalizing for equal protein concentration, tissue lysates were resuspended in non-reducing sample buffer and separated by SDS-PAGE. Following overnight transfer of the proteins onto nitrocellulose or PVDF membranes, the membranes were blocked with 5% BSA (w/v) in wash buffer supplemented with 2 mM CaCl$_2$ and probed with the following antibodies overnight at 4 °C: LDLR (SCBT #sc-18823; 1:200). After primary antibody incubation, membranes were washed and incubated with fluorescently labeled (Invitrogen #A10043, #A21057; 1:5000) or HRP-conjugated secondary antibodies (Cell Signaling Technologies #7074, #7076; 1: 5000) for 1 h at RT. Protein bands were visualized using the Odyssey Infrared Imaging System (LI-COR Biotechnology) or enhanced chemiluminescence (Pierce). Densitometry analysis of the gels was carried out using ImageJ software from the NIH (http://rsbweb.nih.gov/ij/).

**MMAB/MVK promoter assays.** A ~1.4 kb insert of the MVK/MMAB promoter was purchased from GeneCopoeia (#HPRM12674-PG02) and subcloned into a PGL3 promoter vector (Promega #E1761) in the forward (MVK promoter) and reverse direction (MMAB promoter) using the SacI linker. The primer sequences used for cloning are listed in Supplementary Table 1. The LDLR promoter was purchased from Addgene (#14940). For overexpression assays, HeLa cells were co-transfected with 0.5 μg of the indicated promoter constructs, 0.01 μg of *Renilla* luciferase reporter plasmid (Promega), and 0.5 μg, 1 or 2 μg of nuclear SREBP2 (nSREBP2, Addgene #26807) or empty vector control (ThermoFisher #V79020) in 12-well plates using Lipofectamine 2000 (Invitrogen #11668019) according to the manufacturer's instructions. After 24 h of transfection, luciferase activity was measured using the Dual-Glo Luciferase Assay System (Promega #E2940). *Renilla* luciferase activity was normalized to the corresponding firefly luciferase activity and plotted as a percentage of the control (cells co-transfected with the corresponding concentration of empty vector control) as previously described[65]. In another set of experiments, Huh7 cells were co-transfected with 0.5 μg of the indicated promoter constructs and 0.01 μg of *Renilla* luciferase reporter plasmid

(Promega) in 12-well plates using Lipofectamine LTX/Plus Reagent (Invitrogen #15338100) according to the manufacturer's instructions. Twelve hours following transfection, cells were incubated in DMEM media containing 20% FBS (v/v) and 60 μg/ml nLDL (to cholesterol enrich) or DMEM containing 10% LPDS (v/v)+ 5 μM statin (to cholesterol deplete). After 24 h of treatment, cells were collected and luciferase activity was measured as described above.

**Total sterol and cholesterol biosynthesis measurements.** Huh7 cells were plated in 100 mm dishes and transfected with either 40 nM of a siRNA against MMAB (siMMAB) or non-silencing control siRNA (NS) in DMEM containing 10% (v/v) LPDS as described above. In another set of experiments, Huh7 cells were plated in 100 mm dishes in DMEM containing 10% (v/v) LPDS and treated with 100 μM propionic acid or vehicle control for 24 h as described above. Following treatment, cells were collected and lipids extracted and analyzed by GC-MS as previously described[66]. "Total sterols" were calculated as the sum of all sterols that could be detected. [1,2-14C]-acetate incorporation into cholesterol was determined in Huh7 cells treated with an siRNA against MMAB (siMMAB) or Huh7 cells treated with 100 μM propionic acid by HPLC as previously described[66,67]. In all experiments, an aliquot of cells was saved to determine MMAB knockdown by qRT-PCR as described above. Any replicate that did not achieve ≥80% knockdown was excluded. Hepatic sterol concentrations in mice were measured as previously described[68].

**Global metabolite measurements.** Untargeted global metabolic profiles were obtained in Huh7 cells and human fibroblasts by Metabolon (Durham, NC). Briefly, Huh7 cells were plated in 100 mm dishes and transfected with either 40 nM of siMMAB or NS control siRNA in DMEM containing 10% (v/v) LPDS as described above. In another set of experiments, WT and *MUT* fibroblasts were plated in 100 mm dishes containing DMEM with 10% (v/v) LPDS for 24 h. Following treatment, cells were washed three times with 1× PBS, snap frozen in liquid nitrogen and stored at –80 °C until subsequent analysis. An aliquot of cells was saved to determine MMAB knockdown by qRT-PCR as described above. Any biological replicate that did not achieve ≥80% knockdown was excluded from subsequent analyses (MMAB_2; MMAB_3).

Cells (~100 μl packet pellet) were extracted and prepared for analysis using Metabolon's standard solvent extraction method. Briefly, sample processing was carried out using an automated MicroLab STAR system (Hamilton Company). Recovery standards were added prior to the first step in the extraction process for QC purposes. Sample preparation was conducted using a proprietary series of organic and aqueous extractions to remove the protein fraction while allowing maximum recovery of small molecules. The extracted samples were split into equal parts for analysis of metabolites using GC-MS and /MS platforms.

*LC-MS.* The LC-MS portion of the platform was based on a Waters ACQUITY UPLC and a Thermo-Finnigan LTQ mass spectrometer, which consisted of an electrospray ionization (ESI) source and linear ion-trap (LIT) mass analyzer. The sample extract was split into two aliquots, dried and then reconstituted in acidic or basic LC-compatible solvents, each of which contained 11 or more injection standards at fixed concentrations. One aliquot was analyzed using acidic positive ion optimized conditions and the other using basic negative ion optimized conditions in two independent injections using separate dedicated columns. Extracts reconstituted in acidic conditions were gradient eluted using water and methanol both containing 0.1% formic acid, while the basic extracts, which also used water/ methanol, contained 6.5 mM ammonium bicarbonate. The MS analysis alternated between MS and data-dependent MS scans using dynamic exclusion.

*GC-MS.* The samples destined for GC-MS analysis were re-dried under vacuum desiccation for a minimum of 24 h prior to being derivatized under dried nitrogen using bistrimethyl-silyl-trifluoroacetamide (BSTFA). The GC column was 5% phenyl and the temperature ramp was from 40 °C to 300 °C in a 16-min period. Samples were analyzed on a Thermo-Finnigan Trace DSQ fast-scanning single-quadrupole mass spectrometer using electron impact ionization. The instrument was tuned and calibrated for mass resolution and mass accuracy on a daily basis. Metabolites were identified by their m/z retention time and through comparison with library entities of purified known standards. The dataset comprised a total of 271 compounds of known identity in human fibroblasts and 273 named compounds in Huh7 cells. Raw and normalized metabolite data for fibroblasts and Huh7 cells can be found in Supplementary Data 4 and 5.

**Targeted metabolite measurements.** Methylmalonic acid (MMA) concentrations were analyzed and quantified by LC-MS/MS as described previously with minor modifications[69]. Briefly, liver (~100 mg) and plasma (50 μl) were homogenized in 1 ml ice-cold methanol:water (1:1) and spiked with 50 μl [d3]-methylmalonic acid (0.2 mM). Cells (~0.3 mg protein) were extracted in 300 μl ice-cold methanol:water (1:1) and spiked with 20 μl [d3]-methylmalonic acid (0.2 mM, Cambridge Isotope Laboratories). After centrifugation at 4 °C, concentrations in the supernatants were determined by LC-MS/MS as previously described[69]. Analysis of MMA was performed using gradient elution on RP-C18 columns (Thermo Scientific: 100 × 4, 5μ) with a gradient (10% A to 25% A) elution of acetonitrile (A)

and aqueous 15 mM ammonium formate (B). MS/MS ESI (AB Sciex QTrap 6500) analysis was by a single reaction monitoring mode as the negative ion transitions of 117/73 and 117/55 for MMA and 120/76 and 120/58 for [d3]-MMA.

Plasma propionic acid concentrations were determined by modification of a procedure previously described for the quantification of plasma short-chain fatty acids[70]. Briefly, plasma (25−100 μl), spiked with 10 μl [d5]-propionic acid (2.5 mM, Sigma) was acidified with 20 μl 1 M HCl followed by the addition of ~0.5 g NaCl and derivatized as the amide of 2,4-difluoraniline by addition of 0.2 ml of 1, 3-dicyclohexylcarbodiimide (0.2 M in toluene) and 0.2 ml 2, 4-difluoroaniline (0.2 M in hexane). After 1 h incubation at RT, 2 ml NaHCO₃ (1 M) was added, and the short-chain fatty acid amides of 2,4-difluoraniline were extracted (2×) into ethyl acetate, dried under nitrogen gas, and redissolved in 50 μl ethyl acetate for GC-MS analysis. Analyses were performed on an Agilent GC (HP6890)-MS (HP5973) with a 12 m HP-1 column, EI ionization and monitoring ions m/z 185−190 (m0 to m5) to determine plasma propionate concentration.

**Animal studies.** C57BL/6J (Wild-type, WT) were purchased from Jackson Laboratories (#000664).

Mice deficient in *Apobec1* and heterozygous for *Ldlr* (*Apobec1*⁻/⁻;*Ldlr*⁻/⁺) were a kind gift from Dr. Daniel Rader (UPenn). All animal maintenance and experiments were performed in accordance with NIH guidelines and were approved by the Institutional Animal Care Use Committee of Yale University School of Medicine. Mice were group housed under controlled temperature (22 ± 2 °C), relative humidity (30−70%) and lighting (12 h light/dark cycle; lights on at 7:00 AM) with free access to water and food. Mice were maintained on a regular chow diet (Harlan Teklad 2018S). For high-fat diet (HFD) and statin studies, 8-week-old WT male mice (*n* = 6 per group) were placed on a chow diet, HFD containing 0.3% cholesterol and 21% (w/w) fat (Dyets, Inc), or chow diet supplemented with 0.005% (w/w) rosuvastatin for 3 weeks[71]. Liver samples were collected as previously described[71] and stored at –80 °C until total RNA and protein was harvested for gene expression analysis. For in vivo inhibition of *Mmab*, 8-week-old male *Apobec1*⁻/⁻;*Ldlr*⁻/⁺ mice fed a Western diet (Research Diets, #D12108C) for 2 months were randomized into two groups: LNA control (*n* = 5) and LNA GapmeR against MMAB (*n* = 5). Mice received i.p. injections of 5 mg/kg LNA control (5′-ACGTCTATACGCCCA-3′) or LNA GapmeR MMAB (5′-GAAGTGCGTCCGGATT-3′) oligonucleotides every 3 days for a total of 2 weeks. Twenty-four hours after the final injection, mice were sacrificed and hepatic gene expression analyzed as described above. In another set of experiments, 8-week-old male C57BL/6J mice fed a chow diet and randomized into two groups: Control ASO (*n* = 6) and MMAB ASO (*n* = 7). Mice received one i.p. injection per week of 50 mg/kg control ASO (5′-ACCATGAGGTTCTCAGGGCT-3′) or MMAB ASO (5′-TCAGCCGTGATCACTGGGAC-3′) for 6 weeks. Forty-eight hours after the final injection, mice were sacrificed and liver samples were collected as above and stored at –80 °C for subsequent analyses. Unless otherwise stated, mice were fasted overnight before in vivo studies.

**HMGCR activity assay.** Microsomal HMGCR activity was assessed as previously described with slight modifications[72]. Briefly, microsomal fractions from mouse liver homogenates were obtained via ultracentrifugation (100,000 × *g* for 60 min). The reaction buffer (0.2 M KCl, 0.16 M potassium phosphate, 0.004 M EDTA, and 0.01 M dithiothreitol) containing microsomal protein (200 μg/ml) and 100 μM NADPH was preincubated at 37 °C for 10 min. The reaction was initiated by adding 50 μM HMG-CoA to the reaction buffer. The decrease in the absorbance at 340 nm was followed for 30 min. A similar procedure was followed for the calculation of HMGCR activity in the presence of simvastatin. The specific activity of HMGCR was calculated by subtracting total enzyme activity (without simvastatin) and simvastatin-resistant activity (with simvastatin) and is represented as nmol/ min/mg of total microsomal proteins (extinction coefficient for NADPH is 6.2 mM/cm).

**Lipoprotein profile and lipid measurements.** Mice were fasted overnight (12−16 h) before blood samples were collected by cardiac puncture and plasma separated by centrifugation. HDL-C was isolated by precipitation of non-HDL cholesterol and both HDL-C fractions and total plasma were stored at –80 °C. Total plasma cholesterol and triglycerides were measured using kits according to the manufacturer's instructions (Wako Pure Chemicals, Tokyo Japan). The lipid distribution in plasma lipoprotein fractions was assessed by fast performance liquid chromatography (FPLC) gel filtration with 2 Superose 6 HR 10/30 columns (Pharmacia Biotech, Uppsala Sweden) as previously described[73]. Plasma ALT and AST were measured by COBAS (Roche Diagnostics).

**Statistics and reproducibility.** Sample sizes for cell culture experiments (at least three biological replicates) were chosen based on extensive in vitro experience in the lab. Animal sample size for each study was chosen based on literature documentation of similar well-characterized experiments[71,74–76]. The number of samples used in each study is listed in the figure legends and main text. For siRNA experiments, knockdown was assessed prior to downstream analyses; any biological replicate that did not achieve ≥80% knockdown was excluded from subsequent analyses. For all other in vitro experiments, no data were excluded. Prior to dosing,

mice were randomized into treatment groups to match body weight at the start of the experiment. No animals were excluded from this study. When possible, blinding was performed during data collection and analysis for in vitro and in vivo experiments.

All data are expressed as mean ± s.e.m. Statistical differences were measured using an unpaired two-sided Student's $t$ test, Welch's two-sample $t$ test or one-way or two-way ANOVA with Bonferroni correction for multiple comparisons when appropriate. Normality was checked using the Shapiro−Wilk normality test. A non-parametric test (Mann−Whitney) was used when data did not pass the normality test. A value of $P \leq 0.05$ was considered statistically significant. Data analysis was performed using GraphPad Prism Software Version 7.0a (GraphPad, San Diego, CA).

**Reporting summary**. Further information on research design is available in the Nature Research Reporting Summary linked to this article.

## Data availability

RNA-seq data and Affymetrix array data that support the findings of this study have been deposited in the Gene Expression Omnibus under accession code GSE169712 and GSE183790, respectively. RNAi screen data that support the findings of this study have been deposited in PubChem with the Assay ID 1671195. Raw metabolomics data for Huh7 cells and human fibroblasts can be found in Supplementary Data 4 and 5. All other data that support the findings of this study are within the article and its Supplementary Information and Source Data files. Source data are provided with this paper.

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

## Acknowledgements

We thank Dr. Chi Yun, Janine Recio, Shauna Katz, and Dr. Ramanuj DasGupta at the NYU RNAi Core for their advice and assistance with the RNAi screen. We thank Rolando Garcia Milian and Elisa Araldi for their assistance with the processing and analysis of RNA-seq data and Affymetrix arrays, respectively. We thank Mario Kahn, John Stack, and Irina Smolgovsky for their excellent technical assistance. This work was supported by grants from the National Institutes of Health (R35HL135820 to C.F.-H.; R01HL105945 and R01HL135012 to Y.S.; K99 HL150234 to L.G.; R01 DK116774, R01 DK119968, and P30 DK045735 to G.I.S.), the American Heart Association (16EIA27550005 to C.F.-H.; 16GRNT26420047 to Y.S. and 17SDG33110002 to N.R.), the Foundation Leducq Transatlantic Network of Excellence in Cardiovascular Research (MIRVAD to C.F.-H.), the American Diabetes Association (1-16-PMF-002 to A.C.-D.) and the Ministerio de Industria y Comercio, Spain (SAF2011-29951 to M.A.L.). CIBERobn is an initiative of ISCIII, Spain. The NYU RNAi core is supported by the Laura and Isaac Perlmutter Cancer Center (NIH/NCI P30CA16087) and the NYSTEM Contract (C026719). NYU Langone's Genome Technology Center is supported by the Cancer Center Support Grant (P30CA016087) at the Laura and Isaac Perlmutter Cancer Center. Figures were created with BioRender.com.

## Author contributions

L.G. and C.F.-H. conceived and designed the study. L.G. optimized and performed the RNAi screen. L.G., G.W.C., A.C.-D., N.R., B.C. and B.M.T. performed experiments and analyzed data. R.G.L., J.G.M., G.I.S., M.A.L., Y.S. and C.F.-H. assisted with experimental design and data interpretation. L.G. and C.F.-H. wrote the manuscript, which was commented on by all authors.

## Competing interests

The authors declare no competing interests.
