## [Peer Review File · Nature Communications]

Reviewer comments, first round -

Reviewer #1 (Remarks to the Author):

This manuscript describes an siRNA screen in Huh7 cells for genes that when silenced influence LDL uptake. A number of positive hits were identified and validated, including MMAB, of interest as this gene is at an early GWAS locus for LDL-C. The data presented suggest that knockdown of MMAB results in increased levels of MMA which inhibit cholesterol synthesis, resulting in activation of SREBP-2 and upregulation of the LDLR.

Specific comments:

1. Given that MMAB was chosen for focused studies in part because it is at an LDL GWAS locus, the authors should also study MVK, also at that locus, in their model system to see if silencing influences LDL uptake, even though it did not meet criteria in the screen.
2. What are the lipids in patients with methylmalonic aciduria?

Reviewer #2 (Remarks to the Author):

The present study by GOEDEKE Leigh et al. establishes a role for MMAB in the regulation of LDLR and cholesterol metabolism.

However, additional information is required to support the demonstration.

In the work presented, the authors developed a functional high-throughput screening test (Genome wide RNAi screen and expression profiling) based on the internalization of fluorescent lipoproteins in liver cells to identify genes involved in the regulation of lipoprotein receptor (LDLR). The results obtained showed that the expression of 7990 genes was affected by nLDL and statin. These results were compared to the genomic wide associated study (GWAS). This led to the identification of 250 hits. The authors then focused on the cobalamin metabolic pathway by investigating the impact of this pathway on LDLR expression and cholesterol biogenesis. The MMAB enzyme was targeted. This enzyme is known to convert VitB12 to adenosylcobalamin located in the mitochondria. From this, the authors showed that the levels of certain fatty acids (propionate, methylmalonate (MMA)) associated with VitB12 metabolism via the mitochondria could regulate LDL receptors and cholesterol biogenesis.

The results obtained provide a body of evidence that tends to show that MMA (which is increased when MMAB is decreased) increases LDLR and decreases cholesterol biogenesis. It is important to specify that this decrease in MMAB-related cholesterol in vitro but also in vivo is significant, without being powerful, which will have to be further discussed. On the other hand, the effects of MMAB on LDLR, both in vitro and in vivo, are important, which is probably due to the screening technique that focuses on this aspect.

The remarks are as follows:

- 1) The high throughput screening method compared to WGAS to identify hits is a powerful, relevant and adapted method on which I have no particular remarks.
- 2) The justification for targeting MMAB according to the results presented in Fig 4 is appropriate. However, the inactivation of MMAB decreases the total cholesterol level per cell relatively little (Fig 4 d and e). Can you justify and discuss?
- 3) The same is true when MUT is inactivated (Fig 5g). Can you justify and discuss?
- 4) It is debatable whether a double inactivation of MMAB and TSM would be more effective in increasing LDLR and decreasing cholesterol biogenesis. This complementary experiment would add weight to the demonstration and would reinforce the importance of the metabolic cobalamin pathway in the metabolism of cholesterol.
- 5) In Fig 4, propionate is used at 100 μ M; justification is given on a bibliographic basis. In humans, according to Dankert et al (1981) (ref 32), the concentration of propionate can reach 100 μ M, but most often it is less than 10 μ M in human plasma? Can you justify this? What would be the effect of 5 μ M propionate (like statin)? What is the concentration of MMA used? Additional data and information are required.

- 6) In Fig 4 a-b, how do you experimentally distinguish between uptake and binding?
- 7) In Fig 6f, the MMA concentrations (1 mM and 10 mM) are very high to inhibit HMGCR (justify and discuss). Can 5 μ M statin be added to this figure?
- 8) In vivo experiments are convincing in terms of profile (reduction of HMGCR activity and increase of SREBP2-mediated gene expression (Fig 7). In MMAB ASO versus CON ASO mice, what about the plasma levels of propionate and MMA? Are they similar or higher in MMAB ASO?
- 9) Furthermore, supplementary Fig 6, despite a decrease in cholesterol precursors in LNA MMAB versus LNA CON; Ffig 6O), no significant difference in cholesterol levels is observed (Fig 6p). Can you explain, discuss ? On the other hand, the VLDL, LDL/IDL and HDL profile is not similar: there would be less LDL in LNA MMAB. This beneficial aspect is to be discussed.
- 10) Are there arguments and/or data supporting that propionate and MMA decreased LDLR and cholesterol biogenesis in mice?
- 11) Figure 8: this figure mainly summarizes data obtained in vitro. Is it possible to revise and complete this figure by integrating the in vivo aspect?

Further results and new points of discussion will allow us to better conclude unequivocally on the role of MMAB in the regulation of LDLR and on the biogenesis of cholesterol.

RESPONSE TO THE REVIEWERS

We would like to thank the reviewers for their careful consideration and their insightful/helpful comments. We are pleased by the overall enthusiasm for this work and are confident that we will be able to address all of the reviewers' primary concerns.

Reviewer #1 (Remarks to the Author):

“This manuscript describes an siRNA screen in Huh7 cells for genes that when silenced influence LDL uptake. A number of positive hits were identified and validated, including MMAB, of interest as this gene is at an early GWAS locus for LDL-C. The data presented suggest that knockdown of MMAB results in increased levels of MMA which inhibit cholesterol synthesis, resulting in activation of SREBP-2 and upregulation of the LDLR”.

Specific comments:

1. *“Given that MMAB was chosen for focused studies in part because it is at an LDL GWAS locus, the authors should also study MVK, also at that locus, in their model system to see if silencing influences LDL uptake, even though it did not meet criteria in the screen”.*

We would like to thank the reviewer for raising this interesting question. To address this point, we have analyzed LDLR protein expression and activity in Huh7 cells transfected with a siRNA against MVK (siMVK) or non-silencing (NS) control siRNA. Our results show that short-term inhibition (24 h) of MVK increases LDLR expression. Surprisingly, we found that prolonged inhibition of MVK (48–60 h) results in significantly reduced LDLR protein expression and Dil-LDL binding and uptake. Collectively, these findings explain why we did not identify MVK as a candidate gene in our initial RNAi screen and suggest that mevalonate and/or other derivatives of mevalonic acid may provide a signal for the repression of *LDLR* expression. In agreement with this observation, previous studies have reported the inhibitory effect of mevalonolactone on hepatic LDLR expression in rats treated with Zaragozic acid, an inhibitor of squalene synthase (Ness et al., 1994). These results have been included in new **Supplementary Fig. 4** and are discussed in the revised version of our manuscript.

2. *“What are the lipids in patients with methylmalonic aciduria?”*

Methylmalonic aciduria (MMA) is an autosomal recessive (1:50,000-100,000 live births) inborn error of metabolism that causes numerous metabolic abnormalities including mitochondrial dysfunction and autophagy impairment. MMA is most frequently caused by mutations in the mitochondrial enzyme, methylmalonyl-CoA mutase (MUT), which results in the impaired isomerization of methylmalonyl-CoA to succinyl-CoA, leading to the accumulation of toxic organic acids (e.g. methylmalonic acid, propionic acid) and causing severe organ dysfunctions and life threatening complications (Gancheva et al., 2020; Luciani and Devuyst, 2020; Manoli et al., 2018). These mitochondrial alterations can result in the accumulation of neutral lipids in the liver (Manoli et al., 2018) and mild/moderate hypertriglyceridemia (Gancheva et al., 2020). Interestingly, some patients with MMA also have low post-prandial circulating HDL-C levels (Gancheva et al., 2020), in support of previous GWAS studies identifying *MMAB* as a susceptibility gene influencing plasma HDL-C levels in humans (Fogarty MP et al. 2010). We have discussed these findings in the revised version of the manuscript.

-

Reviewer #2 (Remarks to the Author):

The present study by GOEDEKE Leigh et al. establishes a role for MMAB in the regulation of LDLR and cholesterol metabolism. However, additional information is required to support the demonstration. In the work presented, the authors developed a functional high-throughput screening test (Genome wide RNAi screen and expression profiling) based on the internalization of fluorescent lipoproteins in liver cells to identify genes involved in the regulation of lipoprotein receptor (LDLR). The results obtained showed that the expression of 7990 genes was affected by nLDL and statin. These results were compared to the genomic wide associated study (GWAS). This led to the identification of 250 hits. The authors then focused on the cobalamin metabolic pathway by investigating the impact of this pathway on LDLR expression and cholesterol biogenesis. The MMAB enzyme was targeted. This enzyme is known to convert VitB12 to adenosylcobalamin located in the mitochondria. From this, the authors showed that the levels of certain fatty acids (propionate, methylmalonate (MMA) associated with VitB12 metabolism via the mitochondria could regulate LDL receptors and cholesterol biogenesis. The results obtained provide a body of evidence that tends to show that MMA (which is increased when MMAB is decreased) increases LDLR and decreases cholesterol biogenesis. It is important to specify that this decrease in MMAB-related cholesterol *in vitro* but also *in vivo* is significant, without being powerful, which will have to be further discussed. On the other hand, the effects of MMAB on LDLR, both *in vitro* and *in vivo*, are important, which is probably due to the screening technique that focuses on this aspect.

The remarks are as follows:

1) ***“The high throughput screening method compared to WGAS to identify hits is a powerful, relevant and adapted method on which I have no particular remarks”.***

We appreciate the reviewer’s positive comments about our high-throughput screening method.

2) ***“The justification for targeting MMAB according to the results presented in Fig 4 is appropriate. However, the inactivation of MMAB decreases the total cholesterol level per cell relatively little (Fig 4 d and e). Can you justify and discuss?”***

We would like to thank the reviewer for his insightful comment. As shown in **Fig. 4d** and **e**, suppression of MMAB significantly reduces total cholesterol levels by 20% and cholesterol biosynthesis by more than 50%. We think that this effect is meaningful given that cellular cholesterol levels are tightly regulated. Indeed, previous studies from the Goldstein and Brown laboratory demonstrated that 5% depletion of cholesterol in the ER was sufficient to induce SREBP2 transport from the ER to the Golgi, cleavage and transcriptional activation of SREBP2-responsive genes including the LDLR (Radhakrishnan et al., 2008). As suggested by the reviewer, a more detailed description of these implications have been added to the discussion section of the revised manuscript.

3) ***“The same is true when MUT is inactivated (Fig 5g). Can you justify and discuss?”***

We thank the reviewer for raising this important point. As shown in **Fig 5h**, MUT fibroblasts show a 50% reduction in total cholesterol content (28546.2 ± 1944 vs 14656.7 ± 70.8 ng cholesterol/mg protein). We realize that this may not be clear due to the scale used to incorporate all of the sterols in one graph and apologize for this confusion. As mentioned above, we think that this 50% reduction in total cholesterol with *MUT* deficiency is meaningful given that cellular cholesterol

levels are tightly regulated. To highlight these changes, we have now included % reduction new **Fig. 5h**.

4) ***“It is debatable whether a double inactivation of MMAB and MUT would be more effective in increasing LDLR and decreasing cholesterol biogenesis. This complementary experiment would add weight to the demonstration and would reinforce the importance of the metabolic cobalamin pathway in the metabolism of cholesterol”.***

As suggested by the reviewer, we have performed additional experiments to assess the specific contribution of silencing MMAB and MUT in regulating LDLR expression. Specifically, we transfected human hepatoma (Huh7) cells with an siRNA against MMAB, MUT or both and assessed LDLR protein levels by Western blotting. As shown in new **Supplementary Fig. 5**, inhibition of MMAB and MUT results in a marked increase in LDLR protein levels. Interestingly, suppression of both genes leads to a modest further increase in LDLR expression compared to cells treated with a siRNA against MMAB or MUT alone, suggesting a potential additional effect of both genes in regulating LDLR expression and reinforcing the importance of the adenosylcobalamin pathway in regulating cholesterol homeostasis. A detailed description of these findings has been added to the revised version of the manuscript.

5) ***“In Fig 4, propionate is used at 100 μ M; justification is given on a bibliographic basis. In humans, according to Dankert et al (1981) (ref 32), the concentration of propionate can reach 100 μ M, but most often it is less than 10 μ M in human plasma? Can you justify this? What would be the effect of 5 μ M propionate (like statin)? What is the concentration of MMA used? Additional data and information are required”.***

We thank the reviewer for this critical observation and suggestion. While the plasma concentration of propionate is often less than 10 μ M in circulation, previous studies have found that propionate can reach levels of 100-200 μ M (average 32-88 μ M) in the portal vein of humans (Dankert *et al* 1981; Cummings, J. H., et al., 1987). Moreover, primary human hepatocytes treated with 0.3–10mM of sodium propionate accumulate intracellular propionyl-CoA levels (~2-20 μ M) similarly those found in primary human hepatocytes from patients with methylmalonic aciduria (M.S. Collado *et al* 2020). Collectively, we believe that these studies justify the concentrations of propionic acid (25- 1000 μ M) used in our initial dose-finding studies. As shown in **Supplementary Fig. 6a**, we found that propionic acid (sodium salt) caused a significant upregulation of Dil-LDL uptake at concentrations of propionic acid as low as 75 μ M. We chose to use 100 μ M propionic acid for all subsequent analyses given that it showed a maximum increase in Dil-LDL activity compared to vehicle-treated cells. Lower doses of propionic acid (<50 μ M) did not significantly alter LDL uptake (**Supplementary Fig. 6a**) or LDLR expression (**Fig. 1 for Reviewer**) in Huh7 cells.

Fig. for Reviewer 1: LDLR protein expression in Huh7 cells incubated in DMEM containing 10% LPDS for 24 h and treated with 5–50 μ M propionic acid (PA) or vehicle control for an additional 24 h. HSP90 was used as a loading control.

To determine what concentrations of MMA to use, we performed a similar dose response experiment where we assessed Dil-LDL uptake in Huh7 cells exposed to varying doses of MMA (10-500 μ M). These levels are comparable to those previously observed in primary human hepatocytes isolated from patients with methylmalonic aciduria (Collado, M. S., et al. 2020). As shown in **Supplementary Fig. 6c**, we found that MMA increased Dil-LDL uptake in Huh7 cells at

concentrations as low as 25 μM , a concentration of MMA previously observed in serum from aged individuals (Gomes *et al* 2020). Given that 50 μM showed the maximum increase in Dil-LDL activity (**Supplementary Fig. 6c**), we chose to use 50 μM for all subsequent *in vitro* studies. As suggested by the reviewer, we have now explained our rationale for these doses in the revised version of the manuscript.

6) “In Fig 4 a-b, how do you experimentally distinguish between uptake and binding?”

To determine LDL binding to the LDLR in the plasma membrane, cells are pre-incubated at 4°C, which is a common method for inhibition of endocytosis, including the internalization of the LDLR (Calvo *et al.*, 1998; Suarez *et al.*, 2004). LDL uptake is performed at 37°C, a temperature that allows endocytosis and recycling of the LDLR. We apologize for this confusion and have further clarified these experimental procedures in the revised material and methods section of the manuscript.

7) “In Fig 6f, the MMA concentrations (1 mM and 10 mM) are very high to inhibit HMGCR (justify and discuss). Can 5 μM statin be added to this Fig.?”

We thank the reviewer for pointing this out and agree that the 1 mM and 10 mM concentrations of MMA used to inhibit HMGCR activity are supraphysiological (see response to point #5). As such, we assessed HMGCR activity in liver microsomes incubated with lower concentrations of MMA (10–75 μM). Importantly we found that doses as low as 50 μM can significantly inhibit HMGCR activity compared to controls (revised **Fig. 6f**).

The HMGCR activity assay is based on the consumption of NADPH by HMGCR, which can be measured by the decrease in absorbance at 340 nm (**Fig. 2a for Reviewer**). Briefly, HMGCR activity was assessed in microsomal protein (200 $\mu\text{g}/\text{mL}$) incubated with 100 μM NADPH and varying concentrations of MMA or PA. The reaction was initiated by the addition of 50 μM HMG-CoA and the decrease in the absorbance at 340 nm was followed for 30 min at 37 °C. A similar procedure was followed for the calculation of HMGCR activity in the presence of 5 μM simvastatin. Because other enzymes can consume NADPH, the specific activity of HMGCR was calculated by subtracting total enzyme activity (without simvastatin) and simvastatin-resistant activity (with simvastatin) and is represented as nmol/min/mg of total microsomal proteins (extinction coefficient for NADPH 6.2 mM/cm). Representative absorbance plots are shown in **Fig. 2b–c for Reviewer**.

Fig. for Reviewer 2: (a) HMGCR converts HMG-CoA into mevalonate, the rate-limiting step in cholesterol biosynthesis. (b–c) HMGCR activity in liver microsomes treated with varying concentrations of MMA (b) or PA (c) as assessed by NADPH consumption (decrease in absorbance at 340 nm over time).

8) “In vivo experiments are convincing in terms of profile (reduction of HMGCR activity and increase of SREBP2-mediated gene expression (Fig 7). In MMAB ASO versus CON ASO mice, what about the plasma levels of proprionate and MMA? Are they similar or higher in MMAB ASO?”

As suggested by the reviewer, we have analyzed plasma levels of propionate and MMA in wild-type mice treated with an antisense inhibitor of MMAB (MMAB ASO) or control ASO (CON ASO). As expected, we found that mice treated with MMAB ASO have significantly higher concentrations of plasma and hepatic MMA levels compared to controls (**Fig 7b–c**). While MMAB ASO-treated mice tended to have higher levels of plasma propionate, this did not reach significance (**Fig. 7b**). These findings have been included and discussed in the revised version of the manuscript.

9) “Furthermore, supplementary Fig 6, despite a decrease in cholesterol precursors in LNA MMAB versus LNA CON; fig 6O), no significant difference in cholesterol levels is observed (Fig 6p). Can you explain, discuss? On the other hand, the VLDL, LDL/IDL and HDL profile is not similar: there would be less LDL in LNA MMAB. This beneficial aspect is to be discussed”.

We thank the reviewer for raising this point. Unfortunately, mouse models have a number of limitations for studying lipoprotein metabolism, including the absence of CETP and higher LDLR affinity for LDL. As a result, most of circulating cholesterol in wild-type mice is carried by HDL particles. To increase circulating levels of LDL, we performed additional studies in *Apobec^{-/-};Ldlr^{+/-}* mice, which display a more LDL-dominant lipoprotein profile (Ibrahim S *et al* 2016; Kassim SH, 2013). As noted by the reviewer, we found reduced plasma LDL cholesterol levels in mice treated with LNA MMAB compared to LNA CON treated mice, which correlates with increased levels of hepatic LDLR (**Supplementary Fig. 8r**). These results have been discussed in greater detail in the revised discussion of the manuscript and are highlighted in revised **Fig. 8b**.

10) “Are there arguments and/or data supporting that propionate and MMA decreased LDLR and cholesterol biogenesis in mice?”

Our data suggest that the accumulation of both MMA and propionate suppress hepatic cholesterol biosynthesis and increases LDLR expression. As shown in new **Fig. 7b** and **c**, hepatic inhibition of MMAB in mice results in the accumulation of MMA ($P<0.05$) and propionate ($P= 0.1$), which results in higher expression of LDLR (**Fig. 7d**). Moreover, Western-diet fed *Apobec^{-/-};Ldlr^{+/-}* mice treated with a LNA against MMAB displayed a reduction in cholesterol biosynthetic intermediates (**Supplementary Fig. 8o**). As a result, levels of circulating LDL in mice treated with LNA MMAB were reduced compared to LNA CON mice (**Supplementary Fig. 8r**).

While the role for propionate in regulating cholesterol homeostasis *in vivo* is mixed (Illman *et al* 1988; Venter *et al* 1990a; Hara H *et al* 1999; Chen *et al* 1984; Berggren AM *et al* 1996), our findings are consistent with previous reports demonstrating a role for propionate in regulating hepatic cholesterol biosynthesis (Hara H *et al* 1999; Chen *et al* 1984) and plasma levels of cholesterol in rodents (Illman *et al* 1988; Hara H *et al* 1999; Chen *et al* 1984) and humans (Amaral *et al* 2003). In particular, Hara *et al* demonstrated that rates of hepatic cholesterol biosynthesis were decreased in rats fed a diet rich in SCFAs compared to fiber-free-fed controls (Hara H *et al* 1999). Interestingly, plasma levels of total cholesterol were also significantly reduced and negatively correlated with levels of portal plasma propionate (~250 μ M) 5 h post-SCFA-feeding (Hara H *et al* 1999). While the role of acetate and butyrate in mediating SCFA-mediated reductions in hepatic cholesterol biosynthesis cannot be ruled out, these results are consistent with our findings and suggest that SCFA-mediated reductions in total plasma cholesterol may be due to SREBP2-mediated upregulation of hepatic LDLR.

Less data exists concerning the role of MMA in regulating cholesterol biosynthesis *in vivo*. In rat cerebral cortex and liver, MMA (5 mM) significantly reduced ¹⁴C-acetate incorporation into total lipids (de Mello CF *et al* 1997; Brusque A, *et al.* 2001). Consistent with this, chronic administration

of MMA (plasma concentrations of ~2 mM) to Wistar rats, markedly reduced plasma triglyceride levels and cerebral myelin content. While these studies are consistent with MMA playing a role in lipid synthesis, no changes were observed in circulating plasma cholesterol levels or total cholesterol levels in the rat cerebellum. Given the differences in lipoprotein metabolism between rodents and humans, further studies are warranted to dissect the effects of MMA administration on hepatic cholesterol biosynthesis, LDLR expression and circulating LDL-C levels *in vivo*. As suggested, we have discussed these findings in the revised version of the manuscript.

11) “**Fig. 8: this Fig. mainly summarizes data obtained in vitro. Is it possible to revise and complete this Fig. by integrating the in vivo aspect?**”

We think that this is a great suggestion and have included an additional scheme that summarizes the model of how suppression of MMAB influences hepatic cholesterol biosynthesis, LDLR expression and circulating LDL-C levels *in vivo* (new **Fig. 8b**).

REFERENCES

- Amaral, L., et al. (1993). Effect of propionate on lipid metabolism in healthy human subjects. Falk Symposium no. 73, Strasbourg, France.
- Berggren, A. M., et al. (1996). "Influence of orally and rectally administered propionate on cholesterol and glucose metabolism in obese rats." Br J Nutr **76**(2): 287-294.
- Brusque A, et al. Chronic postnatal administration of methylmalonic acid provokes a decrease of myelin content and ganglioside N-acetylneuraminic acid concentration in cerebrum of young rats. Braz J Med Biol Res **34**, 227-231 (2001).
- Calvo, D., Gomez-Coronado, D., Suarez, Y., Lasuncion, M.A., and Vega, M.A. (1998). Human CD36 is a high affinity receptor for the native lipoproteins HDL, LDL, and VLDL. J Lipid Res **39**, 777-788.
- Chen WJ, Anderson JW, Jennings D. Propionate may mediate the hypocholesterolemic effects of certain soluble plant fibers in cholesterol-fed rats. Proc Soc Exp Biol Med **175**, 215-218 (1984).
- Collado, M. S., et al. (2020). "Biochemical and anaplerotic applications of in vitro models of propionic acidemia and methylmalonic acidemia using patient-derived primary hepatocytes." Mol Genet Metab **130**(3): 183-196.
- Cummings, J. H., et al. (1987). "Short chain fatty acids in human large intestine, portal, hepatic and venous blood." Gut **28**(10): 1221-1227.
- Dankert J, Zijlstra JB, Wolthers BG. Volatile fatty acids in human peripheral and portal blood: quantitative determination vacuum distillation and gas chromatography. Clin Chim Acta **110**, 301-307 (1981).
- Fogarty MP, Xiao R, Prokunina-Olsson L, Scott LJ, Mohlke KL. Allelic expression imbalance at high-density lipoprotein cholesterol locus MMAB-MVK. Hum Mol Genet. 2010;19(10):1921–9.
- Gancheva, S., Caspari, D., Bierwagen, A., Jelenik, T., Caprio, S., Santoro, N., Rothe, M., Markgraf, D.F., Herebian, D., Hwang, J.H., et al. (2020). Cardiometabolic risk factor clustering in patients with deficient branched-chain amino acid catabolism: A case-control study. J Inherit Metab Dis **43**, 981-993.
- Gomes, A. P., et al. (2020). "Age-induced accumulation of methylmalonic acid promotes tumour progression." Nature **585**(7824): 283-287.
- Hara, H., et al. (1999). "Short-chain fatty acids suppress cholesterol synthesis in rat liver and intestine." J Nutr **129**(5): 942-948.
- Ibrahim, S., et al. (2016). "Stable liver-specific expression of human IDOL in humanized mice raises plasma cholesterol." Cardiovasc Res **110**(1): 23-29.

Illman, R. J., et al. (1988). "Hypocholesterolaemic effects of dietary propionate: studies in whole animals and perfused rat liver." *Ann Nutr Metab* **32**(2): 95-107.

Kassim, S. H., et al. (2013). "Adeno-associated virus serotype 8 gene therapy leads to significant lowering of plasma cholesterol levels in humanized mouse models of homozygous and heterozygous familial hypercholesterolemia." *Hum Gene Ther* **24**(1): 19-26.

Lin Y, Vonk RJ, Slooff MJ, Kuipers F, Smit MJ. Differences in propionate-induced inhibition of cholesterol and triacylglycerol synthesis between human and rat hepatocytes in primary culture. *Br J Nutr* **74**, 197-207 (1995).

Luciani, A., and Devuyst, O. (2020). Methylmalonyl acidemia: from mitochondrial metabolism to defective mitophagy and disease. *Autophagy* **16**, 1159-1161.

Manoli, I., Sysol, J.R., Epping, M.W., Li, L., Wang, C., Sloan, J.L., Pass, A., Gagne, J., Ktena, Y.P., Li, L., et al. (2018). FGF21 underlies a hormetic response to metabolic stress in methylmalonic acidemia. *JCI Insight* **3**.

de Mello CF, Rubin M, Coelho JC, Wajner M & Souza DG (1997). Effect of methylmalonate and propionate on [³H] glutamate binding, adenylylase activity and lipid biosynthesis in cerebral cortex of rats. *Biochemical and Molecular Biology International*, **42**: 1143-1150.

Ness, G.C., Zhao, Z., and Keller, R.K. (1994). Effect of squalene synthase inhibition on the expression of hepatic cholesterol biosynthetic enzymes, LDL receptor, and cholesterol 7 alpha hydroxylase. *Arch Biochem Biophys* **311**, 277-285.

Radhakrishnan, A., Goldstein, J.L., McDonald, J.G., and Brown, M.S. (2008). Switch-like control of SREBP-2 transport triggered by small changes in ER cholesterol: a delicate balance. *Cell Metab* **8**, 512-521.

Suarez, Y., Fernandez, C., Gomez-Coronado, D., Ferruelo, A.J., Davalos, A., Martinez-Botas, J., and Lasuncion, M.A. (2004). Synergistic upregulation of low-density lipoprotein receptor activity by tamoxifen and lovastatin. *Cardiovasc Res* **64**, 346-355.

Venter, C. S., et al. (1990). "Effects of dietary propionate on carbohydrate and lipid metabolism in healthy volunteers." *Am J Gastroenterol* **85**(5): 549-553.

Reviewer comments, second round -

Reviewer #1 (Remarks to the Author):

none

Reviewer #2 (Remarks to the Author):

The authors have taken into account my different remarks. The answers given are appropriate. The experiments are convincing.

The data clearly establish that the level of MMAB is modulated by cellular cholesterol levels through SREBP2. It is also clear that knockdown of MMAB decreases intracellular cholesterol levels and augments SREBP2-mediated gene expression and LDL-cholesterol uptake in human and mouse hepatic cell lines. Moreover, in mice treated with antisense inhibitors of MMAB, a significant reduction in hepatic HMGCR activity, hepatic sterol content and increased expression of SREBP2-mediated genes was observed.

The results obtained are perfectly summarised in Figure 8.

RESPONSE TO THE REVIEWERS

We would like to thank the reviewers for their careful consideration of this manuscript and their insightful comments which have substantially improved this manuscript

Reviewer #1 (Remarks to the Author):

none

Reviewer #2 (Remarks to the Author):

“The authors have taken into account my different remarks. The answers given are appropriate. The experiments are convincing. The data clearly establish that the level of MMAB is modulated by cellular cholesterol levels through SREBP2. It is also clear that knockdown of MMAB decreases intracellular cholesterol levels and augments SREBP2-mediated gene expression and LDL-cholesterol uptake in human and mouse hepatic cell lines. Moreover, in mice treated with antisense inhibitors of MMAB, a significant reduction in hepatic HMGCR activity, hepatic sterol content and increased expression of SREBP2-mediated genes was observed. The results obtained are perfectly summarized in Figure 8.”

We thank the reviewer for his/her helpful comments and are pleased that we were able to appropriately address his/her concerns.